# From Ambiguity to Verdict: A Semiotic-Grounded Multi-Perspective Agent for LLM Logical Reasoning

## Abstract

Logical reasoning is a fundamental capability of large language models (LLMs). However, existing studies largely overlook the interplay between *logical complexity* and *semantic complexity*, resulting in methods that struggle to address challenging scenarios involving abstract propositions, ambiguous contexts, and conflicting stances, which are central to human reasoning. We propose **LogicAgent**, a semiotic-square–guided framework that jointly addresses these two axes of difficulty. The semiotic square provides a principled structure for multi-perspective semantic analysis, and LogicAgent integrates automated deduction with reflective verification to manage logical complexity across deeper reasoning chains. To support evaluation under these conditions, we introduce **RepublicQA**, a benchmark that couples semantic complexity with logical depth. RepublicQA reaches *college-level semantic difficulty (FKGL 11.94)*, contains philosophically grounded abstract propositions with systematically constructed contrary and contradictory forms, and offers the most semantically rich setting for assessing logical reasoning in LLMs. Experiments demonstrate that LogicAgent achieves state-of-the-art performance on RepublicQA, with a 6.25% average gain over strong baselines, and generalizes effectively to mainstream logical reasoning benchmarks including ProntoQA, ProofWriter, FOLIO, and ProverQA, achieving an additional 7.05% average gain. These results highlight the strong effectiveness of our semiotic-grounded multi-perspective reasoning in boosting LLMs' logical performance.

## 1 Introduction

Logical reasoning (Smith, 2003) plays a central role in human cognition, enabling structured transitions from ambiguous inputs to definitive conclusions. In AI (Cohen et al., 2020; Vaswani et al., 2017), it underpins tasks such as commonsense reasoning (Wang et al., 2024), mathematical proof (Wang et al., 2023; Eisner et al., 2024; Gao et al., 2023), and philosophical thinking (Paul, 2013). However, robust logical reasoning in natural language remains challenging due to (1) *semantic complexity* (Tuggy, 1993), where expressions admit multiple interpretations or surface forms, and (2) *logical complexity* (Gibson, 1998), which requires reasoning over contextual premises and semantic interactions (Zhang et al., 2020; Ding et al., 2024).

Recent approaches can be grouped into three categories: (1) **Linear Reasoning (LR)** methods, such as Naive Prompting and Chain-of-Thought (Wei et al., 2022); (2) **Aggregative**

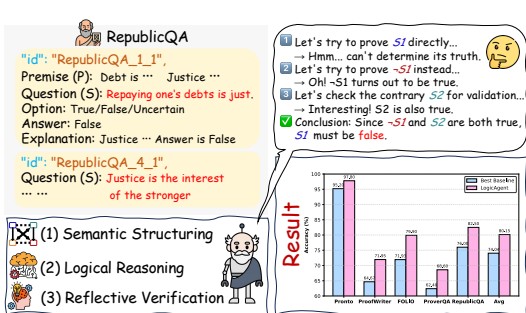

Figure 1: Overview of **LogicAgent** and the proposed **RepublicQA** benchmark. **(Top-left)** RepublicQA features abstract, philosophical propositions from *Plato's Republic* with diverse contextual premises, enabling multiple semantic interpretations. **(Bottom-left)** LogicAgent consists of three stages. **(Top-right)** A multi-step reasoning process explores contraries and contradictions when S1 is indeterminate. **(Bottom-right)** LogicAgent outperforms strong baselines across four benchmarks, demonstrating robust generalization in symbolic, multi-perspective reasoning.

**Reasoning (AR)** approaches (Ryu et al., 2024; Zhang et al., 2023; Yao et al., 2023; Sun et al., 2024) that combine multiple reasoning trajectories; and (3) **Symbolic Reasoning (SR)** frameworks (Yang et al., 2023; Xu et al., 2024a;b) that integrate LLMs (Achiam et al., 2023; Guo et al., 2025; Zhao et al., 2023b) with explicit symbolic modules (Pan et al., 2023). Although effective, these methods remain centered on logical structure and give limited attention to semantic complexity, often assuming clean predicates and unambiguous contexts. This neglects how abstraction, conflicting stances, and contextual ambiguity interact with logical reasoning, limiting performance when semantic and logical complexity jointly shape reasoning (Lago, 2009).

To capture the complex semantics and deep logical relations, we draw inspiration from ***Greimas' Semiotic Square*** (Greimas et al., 1982; Greimas, 1987; 1988), a structuralist framework that extends binary oppositions into a four-part structure. It encompasses both *contraries* (e.g., $S_1$ vs. $S_2$, which cannot both be true but may both be false under non-empty domains) and *contradictions* (e.g., $S_1$ vs. $\neg S_1$, which cannot both be true or false). We migrate this semantic framework into classical FOL with additional constraints, enabling structured multi-perspective reasoning that captures both complex semantics and deep abstract logical relations.

Motivated by agent-based paradigms (Hong et al., 2023), we propose **LogicAgent**, a semiotic-square-guided reasoning framework that automates multi-perspective deduction through a three-stage pipeline: (1) ***Semantic Structuring Stage*** constructs a semiotic square to generate perspective variants of a proposition, including its contradiction and contrary, laying the foundation for multi-perspective reasoning and reflection. (2) ***Logical Reasoning Stage*** formalizes the contextual premises and performs symbolic deduction along both the original and contradiction paths. (3) ***Reflective Verification Stage*** assesses the reasoning trajectory through logic-aware reflection and revises conclusions when inconsistencies arise. This design enables LogicAgent to emulate human-like reasoning while systematically addressing semantic ambiguity and logical complexity.

To rigorously evaluate how semantic complexity interacts with logical reasoning, we introduce **RepublicQA**, a benchmark grounded in classical philosophical concepts and annotated through multi-stage, cross-validated human review. Existing reasoning benchmarks such as ProofWriter (Tafjord et al., 2020), ProntoQA (Saparov & He, 2022), FOLIO (Han et al., 2022), and ProverQA (Qi et al., 2025) are largely template-based and focus primarily on logical structure, offering limited semantic depth and little coverage of the ways semantic ambiguity, abstraction, or opposing stances influence logical inference. In contrast, RepublicQA captures semantic complexity through abstract philosophical content and logical complexity through systematically organized contrary and contradictory relations. It also exceeds existing benchmarks across all five semantic complexity indicators, reaching a college-level reading difficulty (FKGL = 11.94) while maintaining the same level of logical reasoning rigor required by prior datasets.

Experimental results show that our method achieves state-of-the-art performance on **RepublicQA**, surpassing strong baselines across different backbone models with an average improvement of 6.25%. To further validate its generalization, we also evaluate LogicAgent on ProntoQA, ProofWriter, FOLIO and ProverQA, where it again achieves superior results with an average gain of 7.05%. These findings confirm the effectiveness of semiotic-grounded multi-perspective reasoning in enhancing LLMs' logical capabilities under ambiguity and conceptual complexity.

## 2 PRELIMINARIES

Reasoning under ambiguity often involves not only binary truth values but also conceptual oppositions such as contraries (just vs. unjust) and contradictions (true vs. false). Classical logical formalisms capture the latter but lack a systematic way to encode the former. To address this gap, we incorporate **Greimas' Semiotic Square** as a bridging device: it provides a structured representation of semantic oppositions, which we then ground in FOL. This integration forms the basis of LogicAgent, allow-

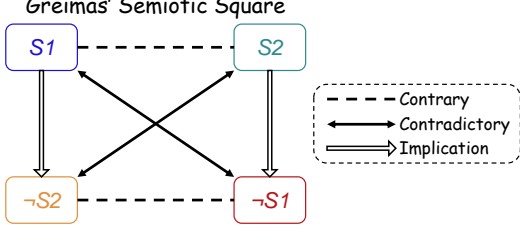

Greimas' Semiotic Square

Figure 2: Greimas' Semiotic Square: illustrating contraries ($S_1$ vs. $S_2$), contradictions ($S_1$ vs. $\neg S_1$, $S_2$ vs. $\neg S_2$), and implications ($S_1 \Rightarrow \neg S_2$, $S_2 \Rightarrow \neg S_1$).

ing it to align natural language semantics with formal logical deduction.

**Greimas' Semiotic Square.** The *Greimas' Semiotic Square* (Greimas et al., 1982) is a foundational construct in structuralist semantics that organizes conceptual contraries and contradictions into a four-element structure, enabling fine-grained reasoning over meaning, opposition, and implication. In our work, we **migrate this semantic structure into the setting of classical FOL**, with additional constraints to ensure logical soundness. Specifically, we introduce an *existential import* check to avoid vacuous truth from material implication, and extend the evaluation space from binary {True, False} to a three-valued scheme {True, False, Uncertain}, which better reflects reasoning under ambiguity.

Let $S_1$ denote a primary proposition. The structure of the semiotic square is illustrated in Figure 2:

- $S_2$: the **contrary** of $S_1$. The relation $S_1 \perp S_2$ implies both cannot be true, but both may be false (subject to non-empty domain constraints).

- $\neg S_1$: the **contradictory** of $S_1$, satisfying the classical law of excluded middle: $S_1 \leftrightarrow \neg\neg S_1$.

- $\neg S_2$: the contradictory of $S_2$.

**Theorem 1** (Semantic Implication Theorem). *If $S_1$ and $S_2$ are contraries, then within the semiotic square the following semantic implications hold:*

$$S_1 \Rightarrow \neg S_2 \quad and \quad S_2 \Rightarrow \neg S_1. \tag{1}$$

*Proof.* Assume $S_1 = $ True. Since $S_1$ and $S_2$ are contraries, we obtain $S_2 = $ False. By the definition of contradiction, $\neg S_2 = $ True. Thus, $S_1 \Rightarrow \neg S_2$. Symmetrically, $S_2 \Rightarrow \neg S_1$.

This structural implication mechanism allows the reasoning agent to verify the coherence of judgments made about a target proposition by leveraging the relational semantics encoded in the square.

## 3 METHODOLOGY

To emulate human-like logical reasoning from multiple perspectives, we propose **LogicAgent**, as shown in Figure 3. The framework consists of three core stages: (1) *Semantic Structuring Stage*, (2) *Logical Reasoning Stage*, and (3) *Reflective Verification Stage*. By integrating semiotic theory with classical logic under additional constraints, LogicAgent enables multi-perspective reasoning and reflection over conceptual structures. Each stage plays a distinct role in transforming linguistic input into structured reasoning: from semantic structuring, to symbolic deduction, and ultimately to reflective verification for consistency.

### 3.1 TASK DEFINITION

Given a set of natural language *Premises* $P = \{p_1, p_2, \ldots, p_n\}$, where each $p_i$ denotes a logical statement, and a *Proposition Q*, the task is to determine the answer of $Q$ with respect to $P$, choosing one of three labels: **True**, **False**, or **Uncertain**. [1]

### 3.2 SEMANTIC STRUCTURING STAGE

Given an input proposition $Q$, this stage first treats it as the primary proposition $S_1$, preserving its original semantic stance. Based on $S_1$, the stage jointly generates its *contradictory* $\neg S_1$, the corresponding *contrary* $S_2$, and the *contradiction of the contrary* $\neg S_2$, each paired with a symbolic representation in FOL.

**Contradictory Construction.** Given a natural-language proposition $S_1$, we first formalize it into a FOL expression. We then negate the entire formula to obtain $\neg S_1$ and simplify it using standard equivalences (quantifier negation, De Morgan, implication, bi-implication), as summarized in Table 1 (column "$S_1$ (simplified)"). Finally, the simplified form is mapped back into natural language, ensuring that $\neg S_1$ is both syntactically valid in FOL and semantically a strict negation of $S_1$.

---

[1]ProntoQA is restricted to the classical two-valued setting (**True/False**), whereas RepublicQA, ProofWriter, FOLIO, and ProverQA additionally include **Uncertain** to handle indeterminate cases.

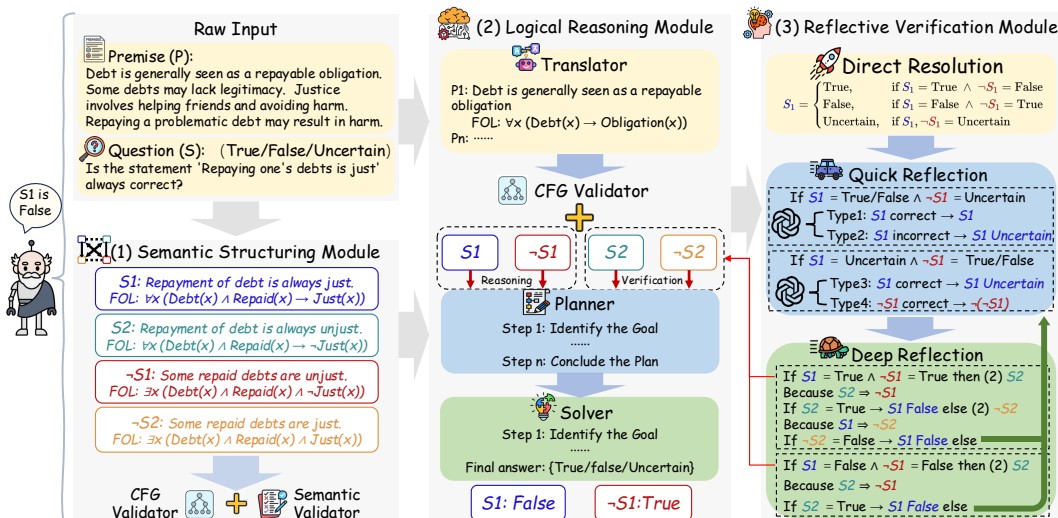

Figure 3: **Overview of the LogicAgent framework.** The agent processes a natural language proposition through three stages. (1) **Semantic Structuring Stage** constructs a Greimas' Semiotic Square, generating four interrelated propositions: the primary proposition $S_1$, its contradiction $\neg S_1$, the contrary $S_2$, and the contradiction of the contrary $\neg S_2$. These are verified for FOL-consistency using a CFG-based parser. (2) **Logical Reasoning Stage** transforms the premises into FOL, plans deductive steps for each proposition, and performs symbolic reasoning to evaluate their answers. (3) **Reflective Verification Stage** adjudicates the final judgment via three procedures: *Direct Resolution*, applied when $S_1$ and $\neg S_1$ offer a contradictory answer; *Quick Reflection*, used when either $S_1$ or $\neg S_1$ is uncertain; and *Deep Reflection*, used when both $S_1$ and $\neg S_1$ yield the same value, requiring further validation through the semiotic implication relations involving $S_2$ and $\neg S_2$.

**Definition 1** (Existential Import Check). Let $P$ be the set of premises defining a model $\mathcal{M}_P = (D_P, I_P)$, where $D_P$ is the domain of discourse and $I_P$ the interpretation function. For a candidate formula $\phi$, its **existential import** under $P$ holds, written $\text{EIC}_P(\phi) = \mathbf{T}$, iff

$$\exists \eta : \text{Free}(\phi) \to D_P \text{ such that } \mathcal{M}_P, \eta \models \text{Ante}(\phi),$$

where $\text{Ante}(\phi)$ denotes the antecedent or quantifier scope of $\phi$. Otherwise $\text{EIC}_P(\phi) = \mathbf{F}$, indicating that $\phi$ is vacuous (e.g., empty domain or unsatisfiable antecedent).

**Lemma 1** (Soundness of Conditional Contrariety). A candidate pair $(S_1, S_2)$ generated under a rule $r$ in Table 1 is a **valid contrary pair** under $P$ iff both satisfy $\text{EIC}_P(S_1) = \text{EIC}_P(S_2) = \mathbf{T}$ and $\mathcal{M}_P \models \neg(S_1 \wedge S_2)$. Pairs failing either condition are excluded from the contrary set.

This establishes that only non-vacuous and mutually unsatisfiable pairs are retained, ensuring that the migration from semiotic structure to FOL preserves logical soundness.

**Contrary Construction.** We adopt the classical definition of contrariety: $S_1$ and $S_2$ cannot both be true but may both be false. Table 1 summarizes six unified rules for constructing contraries and contradictories. For **strict** forms, symbolic transformation directly yields valid contraries. For **conditional** forms, candidates $S_2$ are first generated (via rules or LLM transformation) and then verified by the *existential import check* (Definition 1) to ensure non-vacuous quantifiers and satisfiable antecedents. Only pairs satisfying $\text{EIC}_P(S_1) = \text{EIC}_P(S_2) = \mathbf{T}$ and $\mathcal{M}_P \models \neg(S_1 \wedge S_2)$ are retained as valid contraries (Lemma 1). For structures beyond these six templates, model-assisted generation with self-supervised validation re-applies Definition 1 to filter logically sound pairs.

**Validation and Verification.** All candidate propositions ($S_1$, $\neg S_1$, $S_2$, $\neg S_2$) are validated through a three-stage pipeline, and only those passing all stages are retained for downstream reasoning:

1. **Truth-table evaluation**: FOL formulas are assigned truth values to check whether relations of contrariety ($S_1$ vs. $S_2$) and contradiction ($S_1$ vs. $\neg S_1$, $S_2$ vs. $\neg S_2$) are satisfied.

2. **CFG-based validation**: A context-free grammar (CFG) checker enforces syntactic correctness of all FOL expressions, guaranteeing well-formedness.

3. **LLM verification**: An LLM confirms **semantic and structural consistency**, ensuring that contraries and contradictories remain faithful to the intended meaning and relevant to the premises.

Table 1: Unified rules for constructing contraries and contradictories. $A, B$ denote arbitrary formulas (possibly nested); $\varphi(x)$ denotes a predicate with variable $x$. $\oplus$ is exclusive-or.

| # | $S_1$ | $\neg S_1$ | $S_2$ | Type | Constraint | EIC Condition |
|---|---|---|---|---|---|---|
| 1 | $\forall x\, \varphi(x)$ | $\neg \forall x\, \varphi(x) \equiv \exists x\, \neg\varphi(x)$ | $\forall x\, \neg\varphi(x)$ | Strict | N/A | Always holds |
| 2 | $A \wedge B$ | $\neg(A \wedge B) \equiv \neg A \vee \neg B$ | $A \wedge \neg B$ | Strict | N/A | Always holds |
| 3 | $A \leftrightarrow B$ | $\neg(A \leftrightarrow B) \equiv A \oplus B$ | $A \leftrightarrow \neg B$ | Strict | N/A | Always holds |
| 4 | $\exists x\, \varphi(x)$ | $\neg \exists x\, \varphi(x) \equiv \forall x\, \neg\varphi(x)$ | $\exists x\, \neg\varphi(x)$ | Conditional | $D = \emptyset$ | $\text{EIC}_P(\varphi(x)) = \mathbf{F}$ |
| 5 | $A \rightarrow B$ | $\neg(A \rightarrow B) \equiv A \wedge \neg B$ | $A \rightarrow \neg B$ | Conditional | $\text{Sat}(A)$ | $\text{EIC}_P(A) = \mathbf{T}$ |
| 6 | $A \vee B$ | $\neg(A \vee B) \equiv \neg A \wedge \neg B$ | $A \vee \neg B$ | Conditional | $A = \mathbf{F}$ | $\text{EIC}_P(B) = \mathbf{T}$ |

### 3.3 LOGICAL REASONING STAGE

This stage comprises three functional units: a ***Translator*** for premise formalization, a ***Planner*** for reasoning path construction, and a ***Solver*** for logical deduction.

**Translator.** The translator converts natural-language premises into FOL. Instead of relying on open-ended prompting, this step is guided by a set of **general mapping conventions** that define how linguistic structures are aligned with logical forms. In particular:

- **Entities** (objects, concepts) $\mapsto$ unary predicates, e.g., $Entity(x)$.
- **Actions or relations** $\mapsto$ binary or $n$-ary predicates, e.g., $Action(a, x)$ or $Relation(y, x)$.
- **Roles or agents** $\mapsto$ unary predicates over individuals, e.g., $Role(y)$.
- **Normative or evaluative properties** (just, good, harmful) $\mapsto$ predicates over actions or states, e.g., $Just(a)$, $Good(x)$.

This mapping schema is benchmark-agnostic and applies uniformly across different benchmarks. Each translated formula is further validated by a CFG parser to ensure syntactic correctness, so even if predicate names differ across benchmarks, the logical structure remains well-formed.

**Planner.** For a selected proposition from the semiotic square (e.g., $S_1$), the planner constructs a reasoning blueprint. It specifies the evaluation goal, selects relevant premises, and determines which reasoning rules (e.g., Modus Ponens, Modus Tollens, Conjunction, Generalization) could be applied. The planner may also outline potential counterexample checks and identify implicit contextual relations that, while not explicitly stated in the premises, are salient within the discourse background. Its output is a structured reasoning trajectory, but without issuing a verdict.

**Solver.** The solver operationalizes the planner's blueprint: it applies the designated reasoning rules to the given premises, performs deductions step by step, and generates intermediate conclusions. During this process, it verifies logical consistency and checks for contradictions or counterexamples. The solver outputs both a transparent reasoning trace and the final classification of the proposition as **True**, **False**, or **Uncertain**.

### 3.4 REFLECTIVE VERIFICATION STAGE

This stage adjudicates the final judgment through a three-stage reflective process that ensures coherence among the answers of the semiotic square's four propositions.

**Direct Resolution.** When $S_1$ and $\neg S_1$ produce complementary verdicts, such as $S_1 = \text{True}$ and $\neg S_1 = \text{False}$, the stage directly adopts the answer of $S_1$ as final. This scenario reflects a decisive and non-contradictory judgment grounded in the strict contradiction relationship between the proposition and its contradictory. The decision rule for this resolution strategy is defined as follows:

$$S_1 = \begin{cases} \text{True}, & \text{if } S_1 = \text{True} \,\wedge\, \neg S_1 = \text{False} \\ \text{False}, & \text{if } S_1 = \text{False} \,\wedge\, \neg S_1 = \text{True} \\ \text{Uncertain}, & \text{if } S_1, \neg S_1 = \text{Uncertain} \end{cases} \tag{2}$$

**Quick Reflection.** When either $S_1$ or its contradictory $\neg S_1$ is labeled as *Uncertain*, the stage triggers quick reflection by forwarding the two verdicts and their reasoning traces into a large language model. The model analyzes the internal consistency of the deduction process and returns a refined judgment based on four reflection types:

Case 1: If $S_1 \in \{T, F\}$, $\neg S_1 = U$     Case 2: If $S_1 = U$, $\neg S_1 \in \{T, F\}$

- **Type 1:** $S_1$ correct $\Rightarrow$ Return $S_1 = S_1$    • **Type 3:** $S_1$ correct $\Rightarrow$ Return $S_1 = U$
- **Type 2:** $S_1$ incorrect $\Rightarrow$ Return $S_1 = U$    • **Type 4:** $\neg S_1$ correct $\Rightarrow$ Return $S_1 = \neg(\neg S_1)$

**Deep Reflection.** When both $S_1$ and its contradictory $\neg S_1$ yield the same verdict (e.g., both True or both False), this creates a contradiction under standard logical assumptions. The stage enters *Deep Reflection* mode, leveraging the structured semantic relations provided by the semiotic square, in particular the implications $S_1 \Rightarrow \neg S_2$ and $S_2 \Rightarrow \neg S_1$, to adjudicate which prediction is more likely to be valid.

Case 1: Both $S_1 = $ True, $\neg S_1 = $ True    $\rightarrow$ Solve $S_2$

- If $S_2 = $ True: since $S_2 \Rightarrow \neg S_1 \rightarrow \neg S_1$ is correct $\rightarrow$ Return $S_1 = $ False
- Else $\rightarrow$ Solve $\neg S_2$
    - If $\neg S_2 = $ False: since $S_1 \Rightarrow \neg S_2 \rightarrow S_1$ is incorrect $\rightarrow$ Return $S_1 = $ False
    - Else: Invoke Quick Reflection

Case 2: Both $S_1 = $ False, $\neg S_1 = $ False    $\rightarrow$ Solve $S_2$

- If $S_2 = $ True: since $S_2 \Rightarrow \neg S_1 \rightarrow \neg S_1$ is incorrect $\rightarrow$ Return $S_1 = $ False
- Else: Invoke Quick Reflection

## 4 REPUBLICQA BENCHMARK

Current benchmarks primarily focus on logical complexity while largely overlooking semantic complexity, resulting in limited coverage of abstraction, contextual ambiguity, and nuanced meaning. To address this gap, we construct **RepublicQA**, a benchmark designed to jointly capture logical depth and semantic breadth reasoning.

**Benchmark Construction.** RepublicQA draws from classical philosophical and ethical traditions that explore justice, morality, agency, and knowledge. These sources, characterized by dialogical inquiry and abstract argumentation, provide naturally ambiguous propositions and opposing stances suitable for evaluating advanced logical and semantic reasoning. We extracted propositions and contextual premises manually, with double annotation by two graduate students to ensure logical and semantic consistency. A detailed description of RepublicQA and other benchmarks can be found in Appendix C.1.

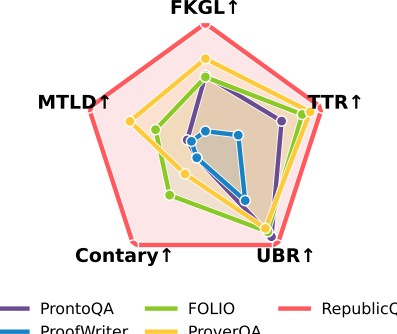

Figure 4: Complexity metrics comparison. Red is our benchmark.

**Complexity Comparison.** RepublicQA introduces abstract propositions with deeper logical dependencies, balanced True/False/Uncertain distributions, and contexts requiring the integration of multiple philosophical concepts. To characterize its complexity, we organize our measurements into three categories: a primary indicator reflecting conceptual difficulty (FKGL), secondary indicators capturing lexical and phrasal variation (TTR, MTLD, UBR), and a supporting indicator concerning the structure of contrary construction. Figure 4 shows that RepublicQA achieves the strongest performance across all five complexity indicators, and its contrary construction patterns far exceed those of existing benchmarks, highlighting its emphasis on abstraction, semantic depth, and non-template reasoning. These properties underscore its suitability for evaluating reasoning under ambiguity and high-level conceptual interactions. Details of these metrics are provided in Appendix D.4, with additional information about RepublicQA in Appendix D.

## 5 EXPERIMENT

### 5.1 SETTINGS

We evaluate our framework on two fronts. First, we assess its performance on our proposed **RepublicQA** benchmark using different baseline models, verifying that the benchmark is broadly applicable for benchmarking logical reasoning. Second, to test the **generalizability** of our method, we conduct evaluations on established logical QA benchmarks, including ProntoQA, ProofWriter, FOLIO, and ProverQA. The experimental setup includes the following components:

**Benchmarks.** We evaluate on four established logical reasoning benchmarks: ProntoQA (Saparov & He, 2022) (5-hop subset), ProofWriter (Tafjord et al., 2020) (depth-5, OWA setting), FOLIO (Han et al., 2022) (full expert-curated split), and ProverQA (Qi et al., 2025) (hard split with 500 examples, 6–9 reasoning steps). Detailed benchmark descriptions are provided in Appendix C.1.

**Baselines.** We compare LogicAgent with five representative baselines: Naive Prompting, Chain-of-Thought (CoT) (Wei et al., 2022), Cumulative reasoning (CR) (Zhang et al., 2023), Tree-of-Thought (ToT) (Yao et al., 2023), Logic-LM (Pan et al., 2023), SymbCoT (Xu et al., 2024b), and Aristotle (Xu et al., 2024a). Detailed baseline introductions are provided in Appendix C.2.

**Model.** For **RepublicQA**, we evaluate with both the locally deployed `qwen2.5:32b` (Yang et al., 2025) and `GPT-4o` (Hurst et al., 2024), ensuring robustness across open and closed source LLMs. For other benchmarks (ProntoQA, ProofWriter, FOLIO, ProverQA), we adopt `qwen2.5:32b` as the base model. In all experiments, the decoding temperature is fixed at 0.

**Symbolic Toolkit.** To verify the syntactic validity of logical forms, we employ the `nltk` (Bird, 2006) library for CFG-based structural checking during the FOL parsing stage.

Table 2: Performance comparison across RepublicQA and other logical reasoning benchmarks. Best results are in **bold**, second-best are underlined.

| Type | Method | RepublicQA | | | Other Benchmarks | | | | |
|------|--------|------------|--------|-------|-----------|---------------|--------|-----------|-------|
| | | Qwen2.5 ↑ | GPT-4o ↑ | Avg ↑ | Pronto ↑ | ProofWriter ↑ | FOLIO ↑ | ProverQA ↑ | Avg ↑ |
| **LR** | Naive | 68.50 | 74.00 | 71.25 | 82.00 | 59.17 | 60.29 | 39.60 | 60.27 |
| | CoT | 72.00 | 75.00 | 73.50 | 92.40 | 63.17 | 68.42 | 47.20 | 67.80 |
| **AR** | CR | 57.00 | 71.00 | 64.00 | 80.20 | 58.33 | 71.57 | 51.80 | 65.48 |
| | ToT | 56.00 | 69.50 | 62.75 | 82.50 | 46.67 | 72.06 | 53.40 | 63.66 |
| **SR** | Logic-LM | 70.00 | 73.50 | 71.75 | 91.89 | 63.82 | 71.93 | 62.40 | 72.51 |
| | SymbCoT | 76.00 | 80.50 | 78.25 | 95.20 | 64.67 | 70.59 | 57.20 | 71.92 |
| **AR+SR** | Aristotle | 74.50 | 82.50 | 78.50 | 94.80 | 63.23 | 68.68 | 56.20 | 70.73 |
| | **LogicAgent** | **82.50** | **87.00** | **84.75** | **97.80** | **71.95** | **79.90** | **68.60** | **79.56** |
| | | (+6.50) | (+4.50) | (+6.25) | (+2.60) | (+7.28) | (+7.97) | (+6.20) | (+7.05) |

### 5.2 COMPARISON WITH SOTA

Table 2 presents the main results from which we can draw several observations.

**Our RepublicQA highlights the unique challenges of semantic ambiguity.** On RepublicQA, Logic-LM performs comparably to the naive baseline, indicating that tool-augmented methods bring little advantage when facing symbolic and semantic ambiguity. In contrast, our LogicAgent achieves the best performance on both Qwen2.5-32B (82.50) and GPT-4o (87.00), surpassing the strongest baseline by an average of 6.25 points. This confirms that RepublicQA effectively stresses reasoning under ambiguity, and that our method is best suited to address these challenges.

**Our LogicAgent generalizes effectively across several mainstream reasoning benchmarks.** LogicAgent achieves an average improvement of 7.05 points over the best baseline, with consistent gains on Pronto (+2.60), ProofWriter (+7.28), FOLIO (+7.97), and ProverQA (+6.20). These results show that the proposed framework transfers robustly beyond RepublicQA and delivers superior performance on diverse logical QA benchmarks.

**Our LogicAgent is particularly effective in multi-hop settings with semantic ambiguity.** Benchmarks such as FOLIO, ProverQA, and RepublicQA require long reasoning chains over context-

sensitive propositions, which often hinder purely neural or template-based approaches. LogicAgent consistently achieves the best performance on these benchmarks, confirming its strength in handling complex reasoning under ambiguity.

## 5.3 ABLATION STUDY

To evaluate the contribution of our method, we conduct three sets of ablation experiments aimed at answering the following key questions:

1. **How effective is each stage in our reasoning framework?** We ablate components to measure their contribution.

2. **What is the impact of FOL representations and natural language descriptions on reasoning performance?** We disable either modality to assess their relative importance.

3. **How do different components affect computational efficiency?** We compare per-sample runtime across ablation settings to study the trade-off between cost and accuracy.

4. **How do semantic and logical complexity interact?** We analyze model performance under varying semantic difficulty and hop depth to make this interplay explicit.

**Q1: Impact of Core Reasoning Stages.** To address **Q1**, we conduct ablation studies on each of the three core stages in our method. Specifically:

- For **Stage 1 (Semantic Structuring)**, we disable the construction of the Greimas semiotic square and retain only the proposition matching the original proposition.

- For **Stage 2 (Logical Reasoning)**, we remove the planning process and directly attempt to solve the proposition without intermediate step generation.

- For **Stage 3 (Reflective Verification)**, we remove both *Quick Reflection* and *Deep Reflection*, and instead apply a rule-based *Direct Resolution* mechanism. Specifically, the model selects the final verdict by combining the base resolution strategy 2 with the supplemental decision rule:

$$S_1 = \begin{cases} S_1, & \text{if } S_1 \neq \text{Uncertain} \wedge \neg S_1 = \text{Uncertain} \\ \neg(\neg S_1), & \text{if } S_1 = \text{Uncertain} \wedge \neg S_1 \neq \text{Uncertain} \\ S_1, & \text{if } S_1 = \neg S_1 \in \{\text{True}, \text{False}\} \end{cases} \quad (3)$$

The ablation results in Table 3 demonstrate that each component in our framework contributes meaningfully to the overall performance. Removing the semiotic square stage causes a substantial decline, with the average accuracy dropping from 75.74 to 67.58 (-8.16). This indicates

Table 3: Ablation results under different configurations.

| Setting | ProofW. | FOLIO | ProverQA | RepublicQA* | Avg |
|---|---|---|---|---|---|
| w/o Square | 65.17 | 72.06 | 56.60 | 76.50 | 67.58 |
| w/o Plan | 62.17 | 69.61 | **75.00** | 72.00 | 69.70 |
| w/o Reflect | 67.50 | 76.12 | 63.40 | 78.50 | 71.38 |
| **Ours** | **71.95** | **79.90** | 68.60 | **82.50** | **75.74** |

that analyzing propositions from multiple semantic perspectives, including contraries and contradictions, is critical for handling complex meanings. Excluding the reflective verification stage results in a smaller decrease to 71.38 (-4.36), suggesting that earlier reasoning stages already yield relatively reliable conclusions. Interestingly, **removing the planning stage improves performance on ProverQA (from 68.60 to 75.00, +6.40)**; however, this removal leads to sharp declines on the other three datasets (-9.78 on ProofWriter, -10.29 on FOLIO, -10.50 on RepublicQA), reducing the average (excluding ProverQA) from 77.78 to 70.94 (-6.84). We observed that ProverQA gold chains typically involve 6–9 reasoning steps, while our planner often generates trajectories exceeding 10 steps. This suggests the existence of a *reasoning complexity threshold*, beyond which over-extended reasoning depth may impair performance, pointing to an important direction for future research.

**Q2: Impact of FOL and Natural Language Inputs.** To assess the individual roles of FOL and natural language inputs, we run ablations that remove either the FOL representations or the natural language statements, as shown in Figure 5a.

**(1) Removing FOL severely impairs symbolic reasoning.** Across all four benchmarks, removing FOL causes large accuracy drops, whereas removing natural language leads to only minor degradation. ProofWriter (71.95 → 33.33), FOLIO (79.90 → 34.31), and ProntoQA (97.80 → 49.00).

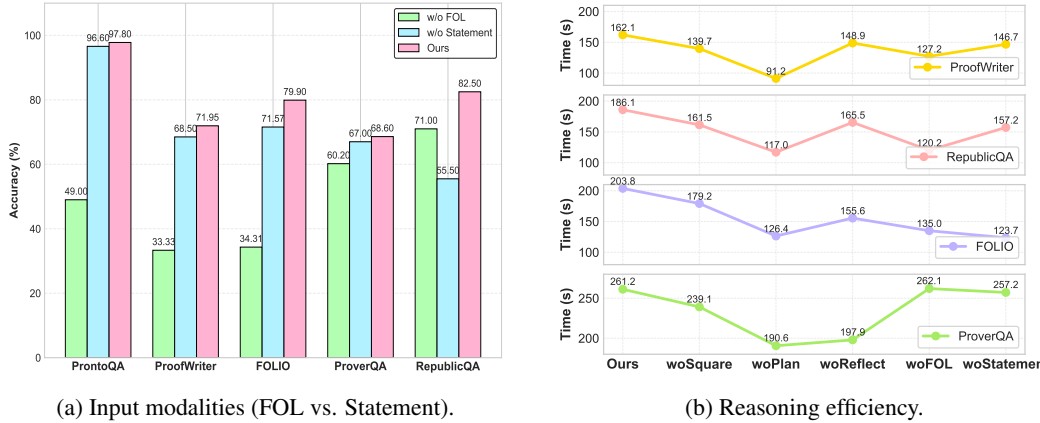

(a) Input modalities (FOL vs. Statement).     (b) Reasoning efficiency.

Figure 5: Ablation studies: (a) input modalities and (b) reasoning efficiency.

ProverQA shows a smaller decrease ($68.60 \rightarrow 60.20$), likely because its natural language descriptions convey relatively clear semantic–logical structure, making it less dependent on explicit FOL representations.

**(2) RepublicQA relies more heavily on natural language semantics.** When natural language input is removed and only FOL is provided, accuracy decreases substantially $82.50 \rightarrow 55.50$, whereas removing FOL leads to a smaller drop to 71.00. This reflects RepublicQA's higher semantic complexity, which arises from implicit assumptions, shifting definitions, and pragmatic dependencies embedded in philosophical discourse that are not fully recoverable through symbolic logic alone.

**(3) Best performance emerges from integrating both modalities.** The highest performance across all five datasets is obtained only when both modalities are used, confirming that FOL provides essential symbolic constraints even in semantically complex settings. Overall, natural language contributes interpretability and contextual grounding, while FOL ensures inferential precision; their integration supports more accurate and robust reasoning across diverse tasks.

**Q3: Impact on Computational Efficiency.** We measure per-sample processing time under various ablation settings on RepublicQA. As shown in Figure 5b, removing planning achieves the largest speedup (-37.1%), reflecting the cost of orchestrating multi-step reasoning. Excluding FOL (woFOL) also reduces processing time by 35.4%, highlighting the overhead of symbolic deduction. Smaller reductions are observed when removing natural language statements (-15.5%) or reflective verification (-9.5%). Similar trends hold for ProofWriter, FOLIO and ProverQA, with planning and FOL as the main bottlenecks. For ProverQA, runtime is instead dominated by reasoning chain length, as removing FOL or statements has little effect.

**Q4: Semantic–Logical Interplay.** As shown in Table 4, we analyze semantic variation and accuracy trends across different hop depths.

Table 4: Semantic Difficulty vs. Logical Depth.

| Hop↑ | FOLIO | | ProofWriter | | ProverQA | | RepublicQA | |
|---|---|---|---|---|---|---|---|---|
| | Acc.↓ | FK | Acc.↓ | FK | Acc.↓ | FK | Acc.↓ | FK↑ |
| 4 | 0.5161 | 6.45 | 0.6029 | 1.65 | 0.5636 | 8.27 | 0.4286 | 12.99 |
| 5 | 0.4474 | 7.72 | 0.4634 | 1.60 | 0.4231 | 8.40 | 0.3125 | 14.68 |
| 6 | 0.3750 | 7.35 | 0.4000 | 1.38 | 0.2500 | 7.40 | 0.0000 | 17.70 |

**(1) Accuracy uniformly declines with greater logical depth.** Across four datasets, accuracy decreases steadily as hop count increases: FOLIO ($0.5161 \rightarrow 0.3750$), ProofWriter ($0.6029 \rightarrow 0.4000$), ProverQA ($0.5636 \rightarrow 0.2500$), and RepublicQA ($0.4286 \rightarrow 0.0000$).

**(2) Prior benchmarks decouple semantic complexity from logical depth.** In FOLIO, ProofWriter, and ProverQA, logical depth increases with hop count, but semantic complexity remains nearly constant. For example, FKGL varies only modestly across hops in FOLIO (6.45–7.72), ProofWriter (1.38–1.65), and ProverQA (7.40–8.40).

**(3) RepublicQA uniquely increases both semantic and logical complexity.** At the same hop level, RepublicQA consistently exhibits higher FKGL yet lower accuracy than all other datasets. For example, around Hop 6, semantic difficulty steadily increases (FKGL $1.38 \rightarrow 7.35 \rightarrow 7.40 \rightarrow 17.70$) while accuracy correspondingly decreases ($0.400 \rightarrow 0.375 \rightarrow 0.250 \rightarrow 0.000$). RepublicQA further shows a clear within-dataset trend: its FKGL rises from 12.99 to 17.70 as hops increase from 4 to 6, while accuracy drops from 0.4286 to 0.0000. This demonstrates that elevated semantic complexity substantially amplifies the difficulty of deep logical reasoning, confirming that RepublicQA is the only benchmark that probes the joint interplay of semantic and logical complexity.

## 6 CONCLUSION

We present **LogicAgent**, a reasoning framework designed to address linguistic ambiguity and complex semantic dependencies through structured, multi-perspective reasoning. Grounded in semiotic principles, LogicAgent transforms natural language into first-order logic representations and performs reasoning in three stages: semantic structuring of contraries and contradictions, FOL deduction along multiple reasoning paths, and reflective verification to refine conclusions. To evaluate reasoning under ambiguity and conceptual variability, we introduce **RepublicQA**, a benchmark derived from philosophical discourse with context-sensitive, semantically rich propositions. Compared with existing benchmarks such as ProntoQA, ProofWriter, FOLIO, and ProverQA, RepublicQA uniquely emphasizes symbolic–semantic alignment under ambiguity, where tool-augmented baselines show limited advantage. Experiments demonstrate that LogicAgent achieves the best performance on RepublicQA and also generalizes strongly across other benchmarks. These findings confirm both the distinct value of RepublicQA and the effectiveness of semiotic-grounded, logic-aware reasoning.

### ETHICS STATEMENT

The RepublicQA benchmark introduced in this paper is derived entirely from publicly available philosophical texts, specifically Plato's *Republic*, which is in the public domain. No human subjects, private data, or sensitive groups are involved, and the dataset poses no immediate ethical or societal risks. All processing steps, including proposition extraction and annotation, were carried out by graduate students following academic standards. For transparency and reproducibility, we plan to release both the code for dataset construction and the RepublicQA benchmark used in our experiments.

### REPRODUCIBILITY STATEMENT

We provide partial code and benchmark with the submission. The complete implementation and the full RepublicQA benchmark will be released at the camera-ready stage to ensure full reproducibility. All experimental settings, model configurations, and benchmark processing details are documented in the main text and Appendix.

### THE USAGE OF LLM

In accordance with ICLR guidelines, we used large language models solely for writing assistance and language refinement.

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

## APPENDIX

This appendix provides supplementary materials, including the related work, formalization in first-order logic (FOL), details of baselines and benchmarks, the construction of RepublicQA, error analysis, case studies, computational efficiency evaluation, and full prompting examples.

## A  RELATED WORK

**LLM-Based Logical Reasoning.** CoT prompting (Wei et al., 2022; Zhang et al., 2025a; Mirzadeh et al., 2024) improves LLM reasoning (Gu & Dao, 2023; Bai et al., 2023; Huang & Chang, 2022) by generating intermediate steps in natural language. Variants such as self-consistency (Wang et al., 2022) and symbolic CoT (Xu et al., 2024b) improve accuracy and interpretability, with SymbCoT showing that symbolic forms enhance logical reasoning. Aristotle (Xu et al., 2024a) further introduces a logic-guided framework using decomposition, search, and resolution, achieving strong results on complex tasks. However, most CoT-based (He et al., 2025; Zhang et al., 2023) and symbolic methods follow linear reasoning paths (Zheng et al., 2024; Zhao et al., 2023a; Sun et al., 2024) and struggle to capture semantic depth, including contraries and contradictions. Other approaches translate LLM outputs into external logic engines (Pan et al., 2023; Ryu et al., 2024), but suffer from brittleness and lack of feedback. In contrast, we propose a fully internal symbolic framework in which the LLM constructs, manipulates, and verifies logical structures. Guided by Greimas' semiotic square (Greimas, 1988), our method enables reasoning over semantically diverse, multi-perspective statements.

**LLM-Powered Agents.** Advances in LLMsSong et al. (2022; 2023); Li et al. (2024); Song et al. (2024); Hu et al. (2025); Song et al. (2025); Ye et al. (2025) have led to agent frameworks (Zhang et al., 2025b; Durante et al., 2024) capable of planning, memory, and multi-step reasoning. Systems such as Generative Agents (Park et al., 2023), AutoAgents (Chen et al., 2023), and MetaGPT (Hong et al., 2023) simulate interactive behavior or coordinate task execution, while others like Code-as-Policies (Liang et al., 2023), Gorilla (Patil et al., 2024), and TaskMatrix (Liang et al., 2024) integrate APIs or GUI actions for real-world applications. Building on the agent paradigm, our approach leverages structured and automated reasoning with symbolic representation and reflective verification to tackle abstract and semantically diverse reasoning tasks. By centering the reasoning process on symbolic representation and multi-perspective analysis, we aim to extend the capabilities of LLMs without additional fine-tuning.

**Benchmarks for Logical Reasoning.** Existing logical reasoning benchmarks (Patel et al., 2024) primarily evaluate formal validity under controlled conditions. **PrOntoQA** (Saparov & He, 2022) is a synthetic relational reasoning benchmark centered on transitivity and set membership in symbolic settings. **ProofWriter** (Tafjord et al., 2020) provides synthetic natural language problems grounded in rule-based microworlds with simplified entities and basic logical connectives. **FOLIO** (Han et al., 2022) contributes natural language scenarios paired with FOL annotations covering everyday commonsense events. **ProverQA** (Qi et al., 2025), generated via the ProverGen pipeline, combines LLM generation with theorem proving to construct verified reasoning chains across multiple difficulty splits. Despite their rigor, these benchmarks focus almost exclusively on *logical form*: their propositions are concrete, unambiguous, and semantically fixed. They omit higher-level semantic complexity, including abstract concepts, contextual variability, and systematically constructed relations such as contraries and contradictions. As a result, they offer limited support for evaluating models' capacity for multi-perspective and contrastive reasoning in semantically rich settings.

## B  FIRST-ORDER LOGIC (FOL)

First-Order Logic (FOL), also known as predicate logic or first-order predicate calculus, is a formal system widely used in mathematics, computer science, philosophy, and linguistics. It extends propositional logic by introducing variables that range over objects in a domain and predicates that describe relationships and properties of these objects. FOL allows us to write general statements involving quantifiers, such as "for all" and "there exists," making it a powerful tool for expressing logical structure and reasoning.

Table 5: Key Syntax Elements in First-Order Logic

| Name | FOL Notation | Explanation |
|---|---|---|
| **Variable** | $x,\ y,\ z$ | Placeholder symbols representing arbitrary elements in the domain of discourse. |
| **Constant** | $a,\ b,\ c$ | Refer to specific, fixed objects in the domain. |
| **Operators (OP)** | $\{\oplus, \vee, \wedge, \rightarrow, \leftrightarrow\}$ | Defines the set of logical connectives used to combine or relate propositions, including exclusive or, or, and, implication, and biconditional. Used in building compound formulas. |
| **Function** | $f(x),\ g(x,y)$ | Maps input objects to an output object; returns a term. |
| **Predicate** | $P(x),\ R(x,y)$ | Express properties or relations; returns true or false. |
| **Negation** | $\neg P(x)$ | Logical NOT: $P(x)$ is not true. |
| **Conjunction** | $P(x) \wedge Q(x)$ | Logical AND: both $P(x)$ and $Q(x)$ must be true. |
| **Disjunction** | $P(x) \vee Q(x)$ | Logical OR: at least one of $P(x)$ or $Q(x)$ must be true. |
| **Implication** | $P(x) \rightarrow Q(x)$ | Logical implication: if $P(x)$ is true, then $Q(x)$ must be true. |
| **Biconditional** | $P(x) \leftrightarrow Q(x)$ | Logical equivalence: $P(x)$ and $Q(x)$ are true or false together. |
| **Universal Quantifier** | $\forall x\ P(x)$ | "For all $x$, $P(x)$ is true" — generalization. |
| **Existential Quantifier** | $\exists x\ P(x)$ | "There exists $x$ such that $P(x)$ is true" — existential claim. |
| **Term** | $x, a, f(a,x)$ | The basic expressions referring to objects (variables, constants, or functions). |
| **Atomic Formula** | $P(a,x)$ | A predicate applied to terms — indivisible logical unit. |
| **Complex Formula** | $\forall x(P(x) \rightarrow Q(f(x)))$ | A formula built from atoms using connectives and quantifiers. |
| **WFF (Well-formed)** | — | A syntactically valid FOL formula interpretable as true or false. |

## B.1 FORMAL SYNTAX AND VALIDATION OF FOL

FOL forms the backbone of our symbolic reasoning pipeline. As shown in Table 5, FOL comprises several syntactic components that define the structure of logical statements, including variables, constants, predicates, logical operators, quantifiers, and term compositions.

**FOL CFG Grammar.** To ensure the well-formedness of FOL expressions, we implement a symbolic parser using the `nltk` library (Bird, 2006). Specifically, we define a context-free grammar (CFG) to support automatic parsing and validation of logical formulas throughout our pipeline:

$$
\begin{aligned}
\text{S} \quad &\to \text{F} \mid \text{Q F} \\
\text{Q} \quad &\to \texttt{QUANT VAR} \mid \texttt{QUANT VAR Q} \\
\text{F} \quad &\to \text{`}\neg\text{' `(' F `)' } \mid \text{ `(' F `)' } \mid \text{ F OP F} \mid \text{L} \\
\text{OP} \quad &\to \text{`}\oplus\text{' } \mid \text{ `}\vee\text{' } \mid \text{ `}\wedge\text{' } \mid \text{ `}\rightarrow\text{' } \mid \text{ `}\leftrightarrow\text{'} \\
\text{L} \quad &\to \text{`}\neg\text{' PRED `(' TERMS `)' } \mid \text{ PRED `(' TERMS `)'} \\
\text{TERMS} \quad &\to \text{TERM} \mid \text{TERM `,' TERMS} \\
\text{TERM} \quad &\to \text{CONST} \mid \text{VAR} \\
\text{QUANT} \quad &\to \text{`}\forall\text{' } \mid \text{ `}\exists\text{'}
\end{aligned}
$$

**Example:** For the rule "$\forall x(\text{Debt}(x) \wedge \text{Repaid}(x) \rightarrow \neg\text{Just}(x))$", the CFG derivation proceeds as follows, as shown in Figure 6:

- `QUANT` $\to$ '$\forall$'

- `PRED` $\to$ 'Debt' | 'Repaid' | 'Just'

- `VAR` $\to$ 'x'

Note that `PRED`, `CONST`, and `VAR` are instantiated dynamically for each example during parsing. This grammar enables symbolic structure checking and forms the foundation for all logic-based components in our agent.

**Syntactic Validation.** We incorporate a rigorous syntactic validation mechanism based on this CFG, serving as a critical quality control step prior to symbolic reasoning. The validator performs structural analysis to ensure:

- **Quantifier Scope Verification:** Ensuring proper binding and scope relationships for universal and existential quantifiers
- **Predicate Structure Validation:** Confirming syntactic correctness of predicate-argument structures
- **Logical Connective Placement:** Verifying appropriate positioning and precedence of logical operators

Only expressions that pass CFG validation are forwarded to the reasoning phase. This ensures the logical integrity of FOL representations derived from natural language and prevents errors caused by malformed logical forms.

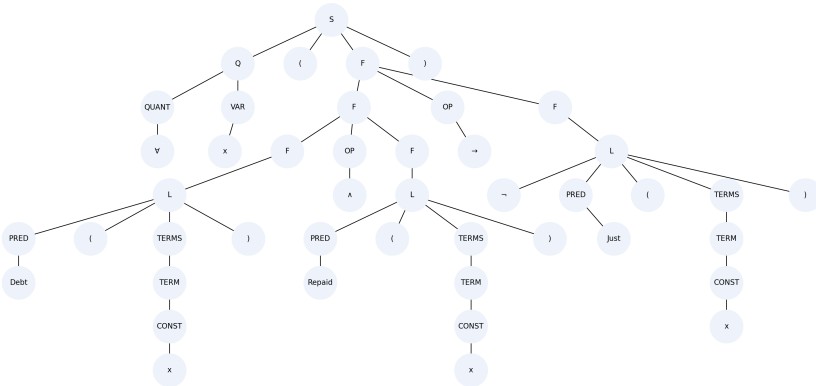

Figure 6: An example CFG parse tree for the FOL rule $\forall x(\text{Debt}(x) \wedge \text{Repaid}(x) \rightarrow \neg \text{Just}(x))$.

## C   BENCHMARKS AND BASELINES

### C.1   BENCHMARKS

**ProntoQA** is a synthetic question-answering benchmark designed to systematically explore the reasoning abilities of language models through formal analysis. The benchmark generates examples with chains-of-thought that describe the reasoning required to answer questions correctly, enabling systematic exploration of LLM reasoning capabilities. The benchmark focuses on fundamental logical relationships and deductive reasoning patterns, providing a controlled environment for assessing model performance on multi-step logical reasoning tasks.

**ProofWriter** is a synthetic benchmark featuring natural language problems that assess systematic neural logical deduction. Developed by the Allen Institute, ProofWriter generates implications, proofs, and natural language reasoning over rulebases of facts and rules under open world assumptions. This benchmark presents complex logical relationships involving combinations of conjunctions and disjunctions, requiring models to perform multi-step deductive reasoning while generating natural language proofs that justify their conclusions. And the context in this benchmark contains more challenging logical relationships such as the combination of "and" and "or."

**FOLIO** is a natural language reasoning benchmark with fol reasoning problems that require models to determine the correctness of conclusions given a world defined by premises. FOLIO aims to ensure high language naturalness and complexity, an abundant vocabulary, and factuality while maintaining high reasoning complexity. It is a high-quality and manually curated benchmark, written by CS undergraduate and graduate students and researchers in academia and industry. To ensure that the conclusions follow the premises logically, all reasoning examples are annotated with FOL formulas. FOLIO represents one of the most challenging logical reasoning benchmarks, combining natural language complexity with the precision of FOL.

**ProverQA** is a high-quality FOL reasoning benchmark created with the ProverGen framework, which combines the generative diversity of LLMs with the rigor of automated theorem proving.

Each instance includes natural language statements, FOL translations, and formally verified reasoning chains. The benchmark is designed to test deductive consistency and the ability to align symbolic and linguistic representations. The dev set contains 1,500 examples evenly divided into easy (1–2 reasoning steps), medium (3–5 steps), and hard (6–9 steps) levels, providing a scalable and systematically validated environment for evaluating logical reasoning under increasing complexity.

**RepublicQA** is a philosophical reasoning benchmark derived from classical works in Western philosophy, including Plato's *Republic* (Plato, 2016), Aristotle's *Metaphysics* (Aristoteles & Apostle, 1966), and the *Nicomachean Ethics* (Irwin et al., 2019). These traditions provide rich discussions of justice, morality, governance, virtue, and knowledge, yielding abstract propositions and structured counterarguments that are well suited for evaluating advanced reasoning. The benchmark presents complex logical problems in which models must judge whether philosophical statements follow from contextual premises. RepublicQA assesses the ability to engage with abstract concepts, moral and ethical reasoning, and classical argumentative patterns while preserving high complexity in both language and inference. Each example reflects foundational questions in Western philosophy and requires reasoning over conditional claims, normative principles, and abstract conceptual relations. To maintain logical consistency, the examples are organized around classical philosophical dialogues and argumentative structures that require multi-step reasoning to determine whether conclusions about justice, virtue, or political order are supported by the given premises.

## C.2 BASELINES

Here we illustrate the details of each baseline used for comparison.

**Naive Prompting.** The model directly receives the question and produces an answer without guidance or intermediate steps. Reasoning is neither encouraged nor structured, making this approach suitable only for simple factual queries.

**Chain-of-Thought (CoT).** CoT prompting elicits step-by-step reasoning before the final answer, improving multi-step reasoning performance by explicitly revealing intermediate steps (Wei et al., 2022).

**Cumulative Reasoning (CR).** CR iteratively refines reasoning across multiple passes. Intermediate outputs from earlier steps serve as inputs to later ones, enabling gradual accumulation and refinement of reasoning (Zhang et al., 2023).

**Tree-of-Thought (ToT).** ToT explores reasoning as a search tree. Instead of a single chain, multiple reasoning paths are generated, evaluated, and pruned, allowing the model to retain only the most promising trajectories (Yao et al., 2023).

**Logic-LM.** Logic-LM translates natural language into first-order logic and applies symbolic solvers for rule-based deduction. This enhances structure and consistency, especially for tasks requiring strict logical validity (Pan et al., 2023).

**SymbCoT.** SymbCoT augments CoT with symbolic representations and logic constraints. Natural language inputs are converted into symbolic forms, and reasoning proceeds under formal logical guidance (Xu et al., 2024b).

**Aristotle.** Aristotle is a logic-complete framework integrating symbolic structures throughout the reasoning pipeline. Its Logical Decomposer, Logical Search Router, and Logical Resolver support structured decomposition, guided search, and contradiction handling, enabling strong performance on complex logical tasks (Xu et al., 2024a).

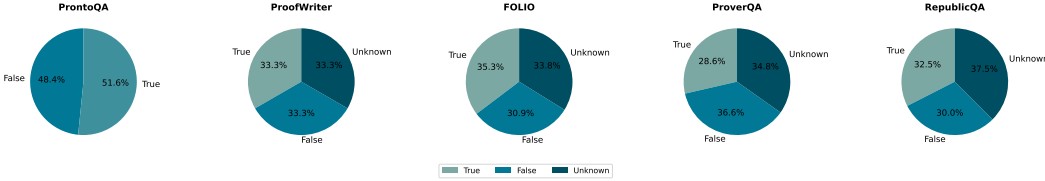

Figure 7: Answer distribution across different benchmarks.

Table 6: Benchmark Comparison: Basic Statistics and Semantic Complexity. **Bold** indicates the best performance and underline indicates the second best.

| Benchmark | Basic Statistics | | | | Semantic Complexity | | | | |
|---|---|---|---|---|---|---|---|---|---|
| | Total | Topics | Vocab | Logic steps | FKGL↑ | TTR↑ | MTLD↑ | UBR↑ | Contrary↑ |
| ProntoQA | 500 | 1 | 69 | 11 | 6.78 | 0.448 | 13.93 | 0.852 | 0.00 |
| FOLIO | 204 | 1 | 1,021 | 0* | 6.62 | 0.569 | 33.54 | 0.805 | 0.30 |
| ProofWriter | 600 | 345 | 61 | 0* | 1.25 | 0.193 | 11.31 | 0.513 | 0.00 |
| ProverQA | 500 | 500 | 2,453 | 25.73 | 8.44 | 0.616 | 34.84 | 0.774 | 0.13 |
| RepublicQA | 600 | 61 | 4,070 | 16.22 | **11.94** | **0.685** | **74.81** | **0.929** | **0.70** |
| | | | | | +41.5% | +11.2% | +114.7% | +9.0% | +133.3% |

*Dataset does not provide explicit reasoning steps.

## D  DETAILS OF OUR REPUBLICQA

### D.1  STATISTICS

**Answer Distribution.** The benchmark exhibits a balanced distribution across three answer categories, as illustrated in Figure 7. The relatively high proportion of "Uncertain" answers (37.5%) reflects the nuanced nature of philosophical reasoning, where definitive conclusions are often difficult to establish.

**Basic Statistics.** The RepublicQA benchmark comprises 600 carefully constructed samples covering 61 unique philosophical topics. Table 6 presents the fundamental statistical characteristics of the benchmark.

### D.2  PHILOSOPHICAL CONCEPTS

RepublicQA is deeply grounded in the thematic structure of Plato's *Republic*, and its philosophical concepts directly shape both the semantic and logical complexity of the benchmark. As shown in Figure 8a, core concepts such as **Justice** (1,308 occurrences), **State** (846), **Soul** (670), **Art** (451), **Knowledge** (242), **Virtue** (206), and **Education** (117) appear frequently throughout the dataset. These abstract and interrelated notions introduce substantial semantic richness and require models to integrate multiple conceptual layers when drawing conclusions. Their interactions create reasoning scenarios that are considerably deeper and more context dependent than those found in benchmarks built around concrete entities or isolated factual predicates.

### D.3  TOPIC MODELING RESULTS

We applied Latent Dirichlet Allocation (LDA) topic modeling to identify thematic structures within the RepublicQA benchmark. Using optimal hyperparameters determined through coherence score validation, we extracted five distinct thematic clusters that capture the core philosophical themes of Plato's Republic:

1. **Political Philosophy**: Encompasses discussions of governance structures, including concepts such as tyrants, wealth distribution, rulers' responsibilities, freedom, and systemic injustice.

2. **Individual Psychology**: Focuses on human nature and development, explaining personal desires, educational processes, external influences on character formation, and individual moral development.

3. **Metaphysics and Epistemology**: Examines questions about knowledge and reality, including the three parts of the soul, how we think rationally, and how different mental faculties work together.

4. **Philosophy of Art and Reality**: Addresses questions of representation and truth, examining concepts of essence, philosophical debate methodology, imitation theory, and the distinction between appearance and reality.

5. **Ethics and Justice Theory**: Examines moral frameworks, investigating justice principles, responses to injustice, ethical rules, power dynamics, and competing moral interests.

These thematic clusters demonstrate the benchmark's comprehensive coverage of Platonic philosophy while maintaining balanced representation across major philosophical domains.

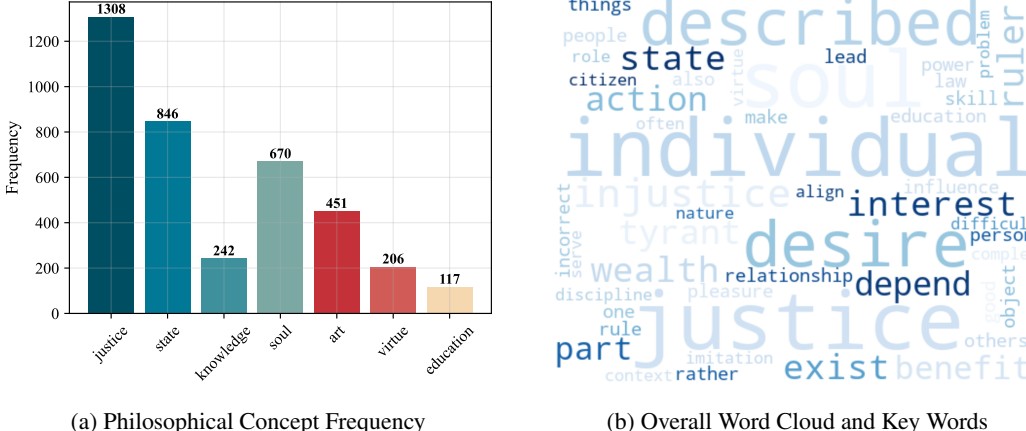

(a) Philosophical Concept Frequency    (b) Overall Word Cloud and Key Words

Figure 8: Analysis of Philosophical Concepts: (a) frequency distribution of concepts, (b) overall word cloud highlighting key terms.

### D.4 SEMANTIC COMPLEXITY ANALYSIS

To assess the semantic complexity of RepublicQA relative to existing reasoning benchmarks, we evaluate all datasets across three complementary dimensions, as summarized in Table 6: (1) **conceptual complexity**, measured by FKGL; (2) **lexical diversity**, captured by TTR, MTLD, and UBR; and (3) **structural contrast**, quantified through the Contrary metric.

**Conceptual Complexity.** We measure sentence-level abstraction using the Flesch–Kincaid Grade Level (FKGL):

$$\text{FKGL} = 0.39 \cdot \frac{N_{\text{words}}}{N_{\text{sentences}}} + 11.8 \cdot \frac{N_{\text{syllables}}}{N_{\text{words}}} - 15.59.$$

Higher scores indicate denser conceptual content and more syntactically demanding propositions.

**Lexical Diversity.** We assess vocabulary and phrasal variation through three complementary measures: Type–Token Ratio (TTR), MTLD for long-span lexical variety, and Unique Bigrams Ratio (UBR) for phrasal diversity. These metrics capture the breadth and stability of semantic expression beyond surface-level repetition.

**Structural Contrast.** To quantify higher-level semantic structure, we use the *Contrary* metric, which measures the presence of systematically constructed contrasting relations within each dataset. Higher values correspond to richer semantic tension and more nuanced relational patterns that require models to integrate multiple, potentially competing interpretations.

**Results.** As shown in Table 6, RepublicQA substantially surpasses existing benchmarks across all dimensions. It requires college-level reading (FKGL = 11.94), exhibits markedly richer lexical diversity (TTR = 0.685), maintains long-span expressive variability (MTLD = 74.81), and achieves a high phrasal diversity (UBR = 0.929). In addition, RepublicQA uniquely incorporates systematically constructed *contrary* relations (Contrary = 0.70), introducing semantic tension and multi-perspective reasoning that are absent from rule-based datasets such as ProntoQA and ProofWriter.

These results establish RepublicQA as a valuable resource for evaluating deep reasoning and generalization in artificial intelligence systems. Figure 8b further illustrates its conceptual landscape through a word cloud of prominent philosophical terms, reinforcing its role as a benchmark for semantically diverse and abstract reasoning tasks.

Table 7: Parallel NL→FOL mappings across difficulty levels for each dataset.

| Dataset | Simple | Medium | Hard |
|---|---|---|---|
| **RepublicQA** | **NL:** Debt repayment is a moral obligation. **FOL:** $\forall a \forall x (\text{Repay}(a, x) \rightarrow \text{MoralObligation}(a))$ | **NL:** Some debts originate from unjust or fraudulent means. **FOL:** $\exists x (\text{Debt}(x) \land (\text{Fraudulent}(x) \lor \text{Unjust}(x)))$ | **NL:** Justice requires helping friends and avoiding harm to innocents. **FOL:** $\forall a (\text{Just}(a) \rightarrow (\forall y (\text{Friend}(y) \rightarrow \text{Beneficial}(a, y)) \land \forall z (\text{Innocent}(z) \rightarrow \neg \text{Harm}(a, z))))$ |
| **ProverQA** | **NL:** Loyal is well-trained. **FOL:** $\text{WellTrained}(loyal)$ | **NL:** If Legend has strong hooves and a powerful gait, he can be a champion. **FOL:** $(\text{StrongHooves}(\ell) \land \text{PowerfulGait}(\ell)) \rightarrow \text{CanBeChampion}(\ell)$ | **NL:** If Legend is competitive, then he has (unique color xor distinctive marking), is good-tempered, not athletic, etc. **FOL:** $\text{Competitive}(\ell) \rightarrow ((\text{UniqueColor}(\ell) \oplus \text{DistinctiveMarking}(\ell)) \land \text{GoodTemperament}(\ell) \land \neg \text{AthleticBuild}(\ell) \dots)$ |
| **ProofWriter** | **NL:** Charlie is kind. **FOL:** $\text{Kind}(charlie, \text{True})$ | **NL:** If someone is quiet and cold, they are smart. **FOL:** $\forall x ((\text{Quiet}(x) \land \text{Cold}(x)) \rightarrow \text{Smart}(x))$ | **NL:** Rough Cold; Cold Smart Red; Red Rough (cyclic reasoning chain). **FOL:** $\text{Rough}(x) \rightarrow \text{Cold}(x); (\text{Cold}(x) \land \text{Smart}(x)) \rightarrow \text{Red}(x); \text{Red}(x) \rightarrow \text{Rough}(x)$ |
| **FOLIO** | **NL:** If people perform, they attend school events. **FOL:** $\forall x (\text{Perform}(x) \rightarrow \text{AttendEngage}(x))$ | **NL:** Inactive people chaperone school dances. **FOL:** $\forall x (\text{InactiveDisinterested}(x) \rightarrow \text{ChaperoneDances}(x))$ | **NL:** Bonnie either attends events as a student, or neither. **FOL:** $\text{AttendEngage}(bonnie) \land \text{StudentSchool}(bonnie) \lor \neg (\text{AttendEngage}(bonnie) \land \text{StudentSchool}(bonnie))$ |

# E  CASE STUDY

We present a representative case from RepublicQA to illustrate how LogicAgent integrates dual-form representations, multi-perspective reasoning, and reflective verification in a unified pipeline. We analyze this example through three key components of our methodology.

**Dual-Form Representation and Semantic Precision**

LogicAgent processes each proposition using both natural language and FOL representations. This dual-form design retains the conceptual richness of natural language while enabling symbolic reasoning under FOL. In the selected case, natural language captures nuanced distinctions (e.g., "justice" vs. "ability"), while FOL clarifies logical scope and reasoning structure:

- **Semantic Preservation:** Contextual meaning is preserved during FOL translation

- **Logical Precision:** Symbolic structure enables explicit reasoning

- **Boundary Clarification:** FOL delineates abstract concepts

To illustrate how LogicAgent handles diverse linguistic phenomena, we additionally provide parallel NL→FOL mappings across simple, medium, and hard cases for all datasets (Table 7), including examples with nested quantifiers and negation.

**Multi-Perspective Reasoning for Robust Evaluation**

To go beyond single-path deduction, LogicAgent constructs a semiotic square for each proposition, enabling reasoning over four semantic positions: $S_1$, $S_2$, $\neg S_1$, and $\neg S_2$. In this case, the system reasons over $S_1$ ("The just man is a thief") and its contradiction $\neg S_1$, revealing a conflict between their conclusions. This multi-perspective reasoning allows:

- **Verification Through Redundancy:** Independent chains confirm or challenge conclusions

- **Error Detection:** Logical inconsistency between perspectives triggers correction

- **Semantic Exploration:** Opposing positions clarify conceptual boundaries

**Planning, Execution, and Reflective Correction**

*Step 1 – Planning:* A 7-step reasoning plan is generated for $S_1$ via semantic decomposition and rule mapping.

*Step 2 – Reasoning Execution:* $S_1$ yields an **Uncertain** result, while $\neg S_1$ concludes **True**, signaling inconsistency.

*Step 3 – Reflective Verification:* The QuickReflection module identifies a Type 4 error (S1 incorrect, ¬S1 correct), attributing it to conceptual confusion between moral capacity and criminal action.

**Final Conclusion:** The system resolves the inconsistency and outputs **False** for the original proposition "The just man turns out to be a thief".

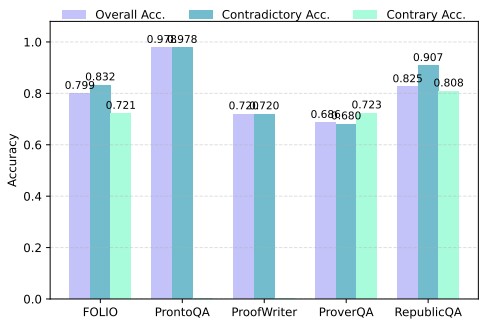

Figure 9: Overall and relation-specific accuracy across datasets.

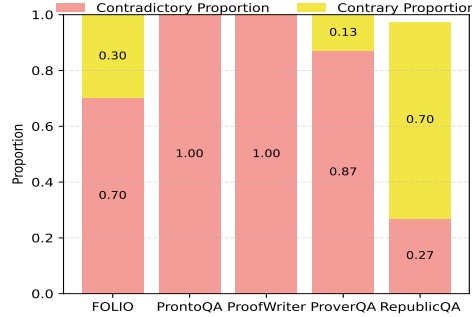

Figure 10: Distribution of contradictory vs. contrary cases across datasets.

## F    ERROR ANALYSIS

Our error analysis highlights four key capabilities required for strong logical reasoning: (1) accurate construction of Greimas' semantic squares, (2) faithful FOL translation, (3) effective planning of reasoning paths, and (4) consistent verification.

**Benchmark Semantic Richness Impact.** As shown in Figure 10, the availability of valid contraries differs substantially across benchmarks. RepublicQA exhibits the richest set of meaningful conceptual contrasts, while FOLIO, ProverQA, ProntoQA, and ProofWriter contain far fewer, which limits opportunities for multi-perspective reasoning. Even within RepublicQA, not all propositions admit well-defined contraries, since constructing them requires resolving semantic ambiguity, aligning abstract concepts, and recovering context-dependent links that are often implicit. Figure 9 further shows that accuracy on contrary cases is consistently lower than overall accuracy, indicating that contrary reasoning remains intrinsically difficult for current models.

**FOL Translation Accuracy.** We observe relatively few FOL parsing errors with Qwen2.5-32B, especially on semantically rich datasets. However, even small translation mistakes can propagate, producing systematic failures despite correct semantic structuring.

**Planning Limitations.** Our framework does not enhance the base model's intrinsic planning ability. When reasoning paths are poorly estimated, semantic analysis alone cannot compensate, especially on long-horizon datasets such as ProverQA. The planner may also over-extend reasoning: while ProverQA typically requires 6–9 steps, our full model often exceeds 10 steps and drops to 68.6% accuracy. In contrast, removing planning (woPlan) keeps trajectories within the expected range and achieves 75% accuracy (Table 3). **These results indicate that reasoning length exhibits a critical threshold beyond which accuracy degrades sharply.**

**Verification Inconsistencies.** Although occurring at extremely low frequencies, our method occasionally exhibits hallucination during the verification phase. In these instances, despite making correct intermediate judgments, the system produces final verdicts that contradict its own reasoning steps, indicating a contradiction between reasoning processes and output generation.

## G  COMPUTATIONAL EFFICIENCY ANALYSIS

We analyze the computational efficiency of LogicAgent from both configuration-level and stage-level perspectives, focusing on processing time and token consumption across core components.

**Time-Accuracy Trade-offs.** As shown in Figure 5b, LogicAgent demonstrates a clear trade-off between accuracy and efficiency. Planning increases computation time by roughly 78% but provides structured, goal-directed trajectories that benefit complex multi-hop reasoning. FOL translation further boosts accuracy through symbolic deduction and consistency checking, though at the cost of substantial latency. In contrast, purely natural language reasoning is faster but lacks the rigor and precision afforded by symbolic structure. These findings highlight the importance of structured reasoning when accuracy and interpretability are required.

**Token Consumption Patterns.** We report token consumption patterns for RepublicQA under three configurations (Table 8). On average, the full setting requires 18.4k tokens, while removing the FOL module (woFOL) reduces usage by 23.5% to 14.1k tokens, with prompt and completion tokens decreasing by 21.7% and 30.1%, respectively. Prompt tokens consistently dominate (75–80% of total), reflecting the heavy contextual demands of multi-hop reasoning. The woStatement setting shows the largest variability, indicating that semantic structuring requirements fluctuate substantially across different philosophical queries.

Table 8: Token Consumption Analysis

| Config | Token Type | Mean | Median |
|---|---|---|---|
| Full | Prompt | 14,402.44 | 13,866.00 |
| | Completion | 3,988.79 | 3,904.50 |
| | Total | 18,391.23 | 18,262.50 |
| woFOL | Prompt | 11,274.76 | 10,586.00 |
| | Completion | 2,788.24 | 2,732.00 |
| | Total | 14,063.00 | 13,405.50 |
| woStatement | Prompt | 13,113.88 | 12,125.00 |
| | Completion | 3,472.03 | 3,309.50 |
| | Total | 16,585.92 | 15,606.00 |

**Stage-Level Timing Breakdown.** To understand intra-system efficiency, we analyze stage-wise processing time for LogicAgent's full configuration on RepublicQA (Table 9). The Logical Reasoning Stage is the dominant computational bottleneck, accounting for 75.1% of total runtime (139.66s). This is attributed to three interacting factors: (1) multi-path execution across $S_1$ and $\neg S_1$, effectively doubling reasoning steps; (2) context-to-FOL translation involving semantic disambiguation and quantifier binding; and (3) execution of detailed plans that enforce step-wise logical progression. The Semantic Structuring Stage, responsible for constructing Greimas' semiotic square, is comparatively efficient (13.6%), as it involves short-form outputs and deterministic linguistic transformations. The Reflective Verification Stage, while occasionally expensive, benefits from its adaptive design. In most cases, it executes lightweight verification; only uncertain or conflicting cases invoke full re-analysis, keeping its average cost low (11.3%).

**Implications.** LogicAgent's modular architecture enables flexible control over computational cost. High-cost components such as planning or FOL translation can be disabled in resource-constrained settings, or selectively activated when tasks demand deeper reasoning.

Table 9: Stage-wise processing time analysis for LogicAgent on RepublicQA

| Stage | Processing Time (s) | Percentage (%) |
|---|---|---|
| Semantic Structuring | 25.31 | 13.6 |
| Logical Reasoning | 139.66 | 75.1 |
| Reflective Verification | 21.08 | 11.3 |
| **Total** | **186.05** | **100.0** |

## H  FULL PROMPTING

Below are detailed prompts used by each component in our reasoning framework, including **Semantic Structuring**, **Translator**, **Planner**, **Solver**, and **Reflective Verification**. Each prompt is designed to elicit structured outputs from the language model to support the overall reasoning pipeline.

---

**Case 1**

**Context:** Some individuals who possess a particular skill may also be able to perform the opposite action of that skill.
......

**Question:** Is the statement 'The just man turns out to be a thief' correct?

**Constructing Semiotic Square**

- **S1 (Assertion):** "The just man turns out to be a thief"
  *FOL Translation:* $\exists x(Just(x) \land Thief(x))$

- **S2 (Contrary):** "The just man turns out to be a guardian"
  *FOL Translation:* $\forall x(Just(x) \rightarrow \neg Guardian(x))$

- **¬S1 (Contradictory):** "The just man does not turn out to be a thief"
  *FOL Translation:* $\forall x(Just(x) \rightarrow \neg Thief(x))$

- **¬S2 (Sub-contrary):** "The just man does not turn out to be a guardian"
  *FOL Translation:* $\exists x(Just(x) \land Guardian(x))$

**Verifying FOL with CFG**

```
[A2] Validation Status: PASSED
- All quantifiers properly scoped
- Predicate structures syntactically correct
- Logical connectives appropriately placed
- Variable bindings consistent
```

**Verifying logical structure** *Logical Structure Verification:*

- **GreimasCoreValid:** true - Core opposition identified

- **GreimasCompleteValid:** true - Complete square constructed

**Translator** The system processes contextual information and formalizes eleven key premises:

1. *Premise 1:* "Some individuals skilled at striking may also be able to defend effectively"
   *FOL:* $\exists y\, (SkillfulAt(Strike, y) \land CanPerform(Defend, y))$

2. *Premises...*

**Multi-Perspective Planning, Reasoning Execution, and Verification**

**Perspective A ($S_1$):**  The system generates a 7-step reasoning plan and executes it.
**Verdict:** $S_1$ reasoning concludes *Uncertain*.

**Perspective B ($\neg S_1$):**  Parallel reasoning is performed for the contradictory proposition.
**Verdict:** $\neg S_1$ reasoning concludes *True*.

**Direct Resolution:**

- **Inconsistency Detected:** $S_1$ = Uncertain and $\neg S_1$ = True violate logical consistency.

- **Trigger:** System enters *Quick Reflection*.

**Quick Reflection Analysis:**

- **S1 Reasoning:** Incorrect.

- **$\neg S_1$ Reasoning:** Correct.

**Reflection Classification:**

- **Type 4 Error:** $S_1$ incorrect, $\neg S_1$ correct with *True* verdict.

- **Resolution Protocol:** Return **False** as final verdict.

- **Error Source:** Conceptual confusion between *ability* and *criminal behavior* in $S_1$ reasoning.

**Final Decision Making:**
Based on QuickReflection analysis revealing conceptual errors in S1 reasoning and confirming the validity of ¬S1 evaluation, the system concludes that the proposition "The just man turns out to be a thief" is **False**.

**Semantic Structuring**

You are a reasoning expert. Your task is to analyze a logical proposition using the Greimas' Semiotic Square framework, which decomposes a proposition into four positions: S1 (original statement), S2 (semantic contrary), ¬S1 (negation of S1), and ¬S2 (negation of S2).

**Core Steps:**

1. **Extract Core Proposition**: If the question asks *"Is the statement 'X' correct?"*, extract $X$ as $S_1$. Preserve original wording exactly.

2. **Identify Semantic Contrary**: Define $S_2$ as a proposition that cannot be true simultaneously with $S_1$, though both may be false. Priority opposition types include:

    - Moral: just vs. unjust, good vs. evil
    - Behavioral: help vs. harm, benefit vs. hurt
    - Authority: obedience vs. independent judgment

3. **Build Semiotic Square**:

    - $S_1$: Original target proposition
    - $S_2$: Semantic contrary to $S_1$
    - $\neg S_1$: Logical negation of $S_1$
    - $\neg S_2$: Logical negation of $S_2$

---

**Example Analysis:**

- **Question**: Is the statement "repaying a debt is always just" correct?

- **Concept A**: just

- **Concept B**: unjust

- $S_1$: Repayment of debt is always just.
  **FOL:** $\forall x \, (Debt(x) \wedge Repaid(x) \rightarrow Just(x))$

- $S_2$: Repayment of debt is always unjust.
  **FOL:** $\forall x \, (Debt(x) \wedge Repaid(x) \rightarrow Unjust(x))$

- ......

- $S_2$ **Type**: Contrary

---

**Output Format (JSON):**

```
{
  "concept\_A": "...",
  "concept\_B": "...",
  "S1": \{"statement": "...", "FOL": "..."\},
  "S2": \{"statement": "...", "FOL": "..."\},
  "not\_S1": \{"statement": "...", "FOL": "..."\},
  "not\_S2": \{"statement": "...", "FOL": "..."\},
}
```

Now analyze the following statement using this framework.
Question: {question}

**Translator**

You are a logical reasoning expert skilled in translating natural language into precise logical structure.

Your task is to extract a list of key **premises** from the following context.

Each premise must be expressed in **two formats**:

1. A concise and accurate **natural-language statement**

2. Its corresponding **First-Order Logic (FOL)** expression written in standard predicate logic

**FOL rules:**

- Logical conjunction of $expr_1$ and $expr_2$:    $expr_1 \land expr_2$
- Logical disjunction of $expr_1$ and $expr_2$:    $expr_1 \lor expr_2$
- Logical exclusive disjunction of $expr_1$ and $expr_2$:    $expr_1 \oplus expr_2$
- Logical negation of $expr_1$:    $\neg expr_1$
- $expr_1$ implies $expr_2$:    $expr_1 \rightarrow expr_2$
- $expr_1$ if and only if $expr_2$:    $expr_1 \leftrightarrow expr_2$
- Logical universal quantification:    $\forall x$
- Logical existential quantification:    $\exists x$

---

**Conventions & Guidelines**

- Use explicit **action variables** (a) for actions like "repaying" or "obeying", and **object variables** (x) for debts, obligations, or rules.

- Use **person or role variables** (y) for entities like people, rulers, citizens, friends.

- Predicates must apply directly to valid entities or actions — never nest predicates:

- Typed variables:
  x $\rightarrow$ debt / obligation / rule
  a $\rightarrow$ action
  y $\rightarrow$ person / social role (e.g., friend, ruler, citizen)

- Focus on extracting premises related to **obligation, justice, causality, moral norms**.

- Quantifiers:
  $\forall$ (for all), $\exists$ (there exists), and treat `Most` / `Typically` as $\forall$ (general statements).

- If the context suggests a causal chain (e.g., *problematic debt $\rightarrow$ harm $\rightarrow$ unjust*), **write each causal link as a separate premise** — do not collapse into a single line.

---

Below is the information you need to deal with right now.
Context:
{context}

---

Return your answer in **exactly** this JSON format:

```
{
  "premises": [
    {
      "statement": "...",
      "FOL": "..."
    }
    ...
  ]
}
```

---

**Planner**

You are a logical reasoning expert.
Your task is to draft a **step-by-step reasoning plan** to determine whether a given logical statement is **true**, **false**, or **uncertain**.
The definition of the three options are:

- **True**: If the premises can infer the question statement under FOL reasoning rule

- **False**: If the premises can infer the negation of the question statement under the FOL reasoning rule

- **Uncertain**: If the premises cannot infer whether the question statement is true or false.

**What to do:**

1. Identify the **goal** (the statement to evaluate).

2. Identify which **premises, rules, or definitions** are relevant.

3. Break down how to **logically connect premises** to reach intermediate reasonings.

4. Organize the reasoning steps clearly and sequentially.

5. End with a **final step: determine whether the statement in the goal is true or false or uncertain**, without making the judgment.

---

Below is an example
**Question:**
"Repaying one's debts is always just.",
"$\forall x$ (Debt(x) $\land$ Repaid(x) $\rightarrow$ Just(x))"
**Premises:**

- Justice involves doing good to friends.
  FOL: $\forall a$ (Just(a) $\rightarrow \forall y$ (Friend(y) $\rightarrow$ Beneficial(a,y)))

- ......

```
{
  "plan": [
    "Step 1: Identify the goal...
    ......
    "Step n: Search for counterexamples...
    "Final Step: Decide whether the premises ...
  ]
}
```

---

Below are the premises and questions you need to derive a plan to solve, please follow the instruction and example aforementioned.
**Input:**
**Question**
{target_statement}
**Premises:**
{premises}

---

**Plan:** Make sure you only derive the plan. Do not solve the question and do not determine the truth value of the conclusion at the planning stage. This plan will be used to help guiding a language model to follow step-by-step. The expected final step in the plan is to determine whether the the conclusion is true/false/uncertain.
Do not solve the question and do not determine the truth value at this stage. Only generate a detailed reasoning plan.

---

**Solver**

The task is to determine whether the value of the conclusion/question is **true/false/uncertain** based on the premises.

You must refer to the following first-order logic reasoning rules when making logical reasoning.

**Input Information:**

1. **Semiotic Square** (The statement you need to reason to judge)
2. **Formal Premises** extracted from the context

Your goal is to evaluate whether the statement in the goal logically follows from the premises. Analyze step-by-step.

Please solve the question step by step. During each step, please indicate what first-order logic reasoning rules you used. Besides, show the reasoning process by the logical operators including but not limited to: ⊕ (either or), ∨ (disjunction), ∧ (conjunction), → (implication), ∀ (universal), ∃ (existential), ¬ (negation), ↔ (equivalence). You can combine natural language and logical operators when doing reasoning.

---

**Definitions:**

- **True**: A statement is "true" if it necessarily follows from the given premises using logical rules.

- **False**: A statement is "false" if it is contradicted by the premises or its negation is logically inferred from them or **if there are counterexamples**.

- **Uncertain**: A statement is "uncertain" if there is insufficient information in the premises to determine its truth value conclusively.

---

**Now analyze input**
**Goal:**
{target_statement}
**Premises:**
{premises}
**Plan:**
{PLAN}

---

**Output JSON Format** (place this at the end, Ensure the JSON is valid (no trailing commas)):

```
{
  "steps": [
    "Step 1: ...",
    "Step 2: ...",
    "...",
    "Final answer: {true/false/uncertain}"
  ],
  "verdict": "True" | "False" | "Uncertain"
}
```

**Reflective Verification**

**Task:** Verify the correctness of the execution in determining the value of the conclusion based on the provided context using first-order logic rules.
**Verification Process:**
**Input Analysis:**
Original Execution: [[EXECUTION]]
**Verification Steps:**

1. **Identify the Goal:** Determine the objective of the original execution.

2. **Evaluate the Premises:** List given premises and their first-order logic representations.

3. **Logical Deduction Analysis:**

   - Analyze S1's reasoning chain.
   - Analyze ¬S1's reasoning chain.
   - Check for logical validity and soundness.

4. **Verdict Justification:** Establish which reasoning is correct.

5. **Classification:** Categorize the case type.

6. **Final Conclusion:** Deliver the verified answer.

**Output Format:**
Conclude with a revised answer using the following JSON structure:

```
{
  "verdict": "True" | "False" | "Uncertain",
  "reason": "Type 1: S1 reasoning correct → Return S1's verdict"|
  "Type 2: S1 incorrect, ¬S1 correct with Uncertain verdict → Return Uncertain"|
  "Type 3: S1 correct with Uncertain verdict → Return Uncertain" |
  "Type 4: S1 incorrect, ¬S1 correct with True verdict → Return False" |
  "Type 5: S1 incorrect, ¬S1 correct with False verdict → Return True" |
  "Type 6: Both S1 and ¬S1 incorrect → Return independently verified result"
}
```

**Verification Execution:**
Original Execution: [[EXECUTION]]
**Verify:**
Please indicate the revised answer at the end using CURLY BRACKETS. The response must be one of:

```
{
  "verdict": "True" | "False" | "Uncertain",
  "reason": "Type 1: S1 reasoning correct → Return S1's verdict"|
  "Type 2: S1 incorrect, ¬S1 correct with Uncertain verdict → Return Uncertain"|
  "Type 3: S1 correct with Uncertain verdict → Return Uncertain" |
  "Type 4: S1 incorrect, ¬S1 correct with True verdict → Return False" |
  "Type 5: S1 incorrect, ¬S1 correct with False verdict → Return True" |
  "Type 6: Both S1 and ¬S1 incorrect → Return independently verified result"
}
```

