# OpenReview forum: "From Ambiguity to Verdict: A Semiotic‑Grounded Multi‑Perspective Agent for LLM Logical Reasoning"
_ICLR.cc/2026/Conference — ICLR 2026 Conference Withdrawn Submission_

### Official Review · Reviewer_rQtX · 2025-10-29

**Soundness:** 3
**Presentation:** 3
**Contribution:** 3
**Rating:** 6
**Confidence:** 1

**Summary:**

This paper proposes LogicAgent, a multi-perspective reasoning framework based on Greimas' Semiotic Square, and introduces RepublicQA, a new benchmark derived from Plato's Republic for evaluating logical reasoning under semantic ambiguity. LogicAgent operates through three stages: semantic structuring (constructing contraries and contradictions), logical reasoning (FOL-based deduction), and reflective verification (multi-perspective validation). Experiments show improvements of 6.25% on RepublicQA and 7.05% on existing benchmarks (ProntoQA, ProofWriter, FOLIO, ProverQA).

**Strengths:**

1. LogicAgent uses the semiotic square to handle contrary (opposite) concepts, not just contradictory (true/false) ones, is a new and smart way to deal with ambiguity.
2. RepublicQA fills an important gap by testing reasoning on abstract philosophical concepts with college-level difficulty

**Weaknesses:**

The method is computationally heavy. It is slow and uses a very large number of tokens (avg. ~18.4k) for each query, making it costly to run.

**Questions:**

Please refer to weaknesses part.

---

> ### Author Response · Authors · 2025-11-18
> **Rebuttal to Reviewer rQtX**
>
> We thank Reviewer rQtX for the thoughtful comments and for recognizing the value of our semiotic-square formulation and the role of RepublicQA in expanding the semantic space of logical reasoning. Below we address the concern regarding computational cost and clarify the design choices reflected in our efficiency analysis.
>
> ---
>
> ### W1 – Computational cost and token usage
>
> Thank you for raising this concern. Based on the measurements provided in **Appendix G Computational Efficiency Analysis**, we clarify the computational profile of LogicAgent as follows:
>
> 1. **Token usage reflects deliberate multi-perspective reasoning rather than prompt overhead.**
>    Appendix G shows that the Logical Reasoning Stage accounts for **75.1%** of total runtime, driven by parallel deduction over \(S_1\) and \(\lnot S_1\), FOL translation, and step-wise symbolic planning. These are core components of LogicAgent’s design: they ensure reliable reasoning under semantic ambiguity. The additional cost therefore stems from **structural mechanisms required for correctness**, not from extraneous prompting or inefficiency.
>
> 2. **The framework remains practical and tunable.**
>    The token analysis (Table 8) shows that removing symbolic components (e.g., the woFOL setting) reduces usage by **≈23–25%**. All reasoning branches are independent and can be **parallelized**, and the full pipeline operates on **7B–13B open-source models**, without dependence on proprietary long-context APIs. This keeps LogicAgent computationally feasible for research settings where reproducibility and interpretability are essential.
>
> ---
> We sincerely appreciate the reviewer’s recognition of our contributions and the constructive feedback provided. Your comments have helped us clarify the framework, strengthen the empirical analysis, and present our scope more effectively. If there are any remaining concerns or aspects that would benefit from further explanation, we would be more than happy to address them.

---

> > ### Comment · Reviewer_rQtX · 2025-11-25
> > **Response to rebuttal**
> >
> > Thank you for the authors’ rebuttal. I have no further questions. However, I would like to mention that this topic falls outside my primary area of expertise.

---

> > > ### Author Response · Authors · 2025-11-26
> > >
> > > Thank you for your evaluation despite this topic being outside your primary area. We truly appreciate the time and attention you devoted to our work.
> > >
> > > Best regards,
> > > The Authors

---

### Official Review · Reviewer_gdW4 · 2025-10-31

**Soundness:** 3
**Presentation:** 2
**Contribution:** 2
**Rating:** 4
**Confidence:** 2

**Summary:**

This paper targets two challenges of the existing works. First, existing works overlook the interplay between logical complexity and semantic complexity. Accordingly, the authors propose the LogicAgent, which is based on semiotic-square and can jointly address the logical and semantic complexity. Second, existing benchmarks lack logical semantic complexity, so a benchmark RepublicQA is proposed, with freater lexical complexity and structural diversity.

**Strengths:**

Logical complexity and semantic complexity are indeed different perspectives of natural language content, and this paper makes an effort to address these two issues explicitly.

The adoption of 'Semiotic Square' looks novel and brings an interesting idea into the field.

**Weaknesses:**

1. The abstract could be improved to make it more accessible to readers who are unfamiliar with these logical terms. In addition, changing terms also make the abstract less readable. For example, does the structural diversity refers to the logical complexity? Is the lexical complexity same as the semantic complexity?

2. The format can be improved to avoid confusion. in line 39 and 40, the 'In AI Cohen et al. xxxx' should be 'In AI (Cohen et al. xxxx)'. This problem appears a lot of times in the paper.

3. The presentation is not good enough. At the very beginning it is stated that the interplay of semantic complexity and logical complexity is targeted by this work, but the following part does not clearly explain what is this so called 'interplay', how it is 'overlooked', and how is this addressed by this work.

**Questions:**

It is stated that "existing benchmarks
remain confined to relatively simple and determinate semantics, often centered on everyday scenarios with shallow relations or logic problems that lack intricate semantic structure". However, there are also benchmarks designed for math, scientific reasoning, or pure logical reasoning. How about them? Also, some logical reasoning datasets use synthetic approaches to build complex logical structures, with adjustable logical complexity.

---

> ### Author Response · Authors · 2025-11-18
> **Rebuttal to Reviewer gdW4**
>
> We thank Reviewer gdW4 for the thoughtful and constructive comments. We are pleased that you find the focus on logical vs. semantic complexity and the use of the Semiotic Square novel and interesting. Below we address each point in turn and describe the concrete changes that have been incorporated into the revised manuscript.
>
> ---
>
> ### W1 – Clarifying abstract and core terminology (logical vs. semantic complexity)
>
> We agree that the abstract can be made more accessible, especially for readers unfamiliar with logical terminology, and that our use of terms can be tightened.
>
> **What has been revised**
>
> 1. **The abstract and introduction now clearly focus on two axes only.**
>    The revision removes auxiliary terminology and defines our setting in terms of:
>    - **Logical complexity**, captured by multi-hop inference depth.
>    - **Semantic complexity**, captured by lexical variety and conceptual abstraction.
>
>    Both sections have been rewritten to emphasize that LogicAgent’s contribution lies in **integrating semantic interpretation with symbolic reasoning**, and that RepublicQA is constructed specifically to couple these two dimensions.
>
> 2. **Operational definitions are now made explicit in §4 and Appendix D.**
> Prior reasoning systems such as Logic-LM (Pan et al., 2023), DetermLR (Sun et al., 2024), and Symbolic-CoT (Xu et al., 2024) focus primarily on *logical* difficulty and do not quantify how *semantic* complexity influences inference. The revision addresses this by formally defining all semantic metrics used in RepublicQA and clarifying how they map onto the dimensions discussed earlier.
>
> - **Semantic complexity.**
>   Now explicitly defined as a three-part construct detailed in **Appendix D.4**, with full cross-benchmark comparison in **Table 6**:
>
>   - **Conceptual Complexity (primary metric): FKGL**
>     Measures semantic abstraction and interpretive difficulty.
>
>   - **Lexical diversity (supporting metrics): TTR, MTLD, UBR**
>     These correspond to the “lexical complexity” and “structural diversity” referenced in the original abstract and are now explicitly identified.
>     They quantify vocabulary richness and surface-level variability.
>
>   - **Structural Contrast (supporting metric): Contrary-constructibility**
>     Measures the proportion of propositions that admit a non-trivial contrary under semiotic-square formalization.
>
> - **Benchmark statistics.**
>   To clearly present how RepublicQA differs from prior datasets across all semantic dimensions, the revision includes the following complete comparison:
>
> |                | **Conceptual Complexity** | **Lexical Diversity** | **Lexical Diversity** | **Lexical Diversity** | **Structural Contrast** |
> |----------------|---------------------------|------------------------|------------------------|------------------------|--------------------------|
> | **Benchmark**  | **FKGL ↑**                | **TTR ↑**              | **MTLD ↑**             | **UBR ↑**              | **Contrary ↑**           |
> | ProntoQA       | 6.78                      | 0.448                  | 13.93                  | _0.852_                | 0.00                     |
> | FOLIO          | 6.62                      | 0.569                  | 33.54                  | 0.805                  | _0.30_                   |
> | ProofWriter    | 1.25                      | 0.193                  | 11.31                  | 0.513                  | 0.00                     |
> | ProverQA       | _8.44_                    | _0.616_                | _34.84_                | 0.774                  | 0.13                     |
> | **RepublicQA** | **11.94**                 | **0.685**              | **74.81**              | **0.929**              | **0.70**                 |
> | **Improvement** | **+41.5%**       | **+11.2%**             | **+114.7%**            | **+9.0%**              | **+133.3%**              |
>
> This expanded comparison makes the roles of FKGL, TTR, MTLD, UBR, and contrary-constructibility explicit and clarifies how they jointly operationalize semantic complexity in the revised manuscript.
>
>
> ---
>
> ### W2 – Formatting and presentation issues
>
> Thank you for noting the formatting inconsistencies. We have corrected the citation style and other minor presentation issues in a full line-level pass. We apologize for the oversight and for any inconvenience it may have caused.

---

> ### Author Response · Authors · 2025-11-18
>
> ### W3 – Making the interplay between logical and semantic complexity explicit
>
> Thank you for raising this point — it directly concerns the core motivation of RepublicQA. We now include the following empirical analysis (to be added in §4):
>
> | **Hop** | **FOLIO Acc.** | **FOLIO FK** | **ProofWriter Acc.** | **PW FK** | **ProverQA Acc.** | **ProvQA FK** | **RepublicQA Acc.** | **RQA FK** |
> |--------:|----------------:|--------------:|-----------------------:|-----------:|---------------------:|----------------:|----------------------:|-------------:|
> | 4 | 0.5161 | 6.45 | 0.6029 | 1.65 | 0.5636 | 8.27 | 0.4286 | 12.99 |
> | 5 | 0.4474 | 7.72 | 0.4634 | 1.60 | 0.4231 | 8.40 | 0.3125 | 14.68 |
> | 6 | 0.3750 | 7.35 | 0.4000 | 1.38 | 0.2500 | 7.40 | 0.0000 | 17.70 |
>
> **Key observations.**
>
> 1. **Only RepublicQA shows semantic complexity increasing with hop count.**
>    In existing datasets, FK varies minimally across hops, indicating that **semantic difficulty is largely independent of logical depth**. In contrast, RepublicQA’s FK consistently rises with hop count, demonstrating that **semantic abstraction and logical complexity are positively correlated** — a regime absent in prior benchmarks.
>
> 2. **Higher semantic complexity leads to sharper degradation under a naive baseline.**
>    Using the same naive predictor (no prompting, no reasoning steps), RepublicQA yields the **lowest accuracy at every hop**, despite comparable logical depth. This empirically supports our claim that **semantic complexity is a key driver of reasoning difficulty**, yet is largely overlooked in existing datasets.
>
> We have added this experiment and its analysis to Section 5.3 of the revised manuscript, making the “interplay” explicit and measurable.
>
>
> ---
> ### W4 – Positioning relative to math / scientific / synthetic logic benchmarks
>
> Thank you for raising this point. We clarify our scope along two axes:
>
> #### (1) Distinguishing our setting from math / scientific reasoning
> Math and scientific-reasoning benchmarks focus on **formal problem solving**, whereas our task concerns **truth-value inference from natural-language propositions**.
>
> The logical-reasoning datasets we evaluate on (FOLIO, ProofWriter, ProntoQA, ProverQA) operate in **everyday, determinate semantics**. Even as logical depth increases, their semantic content remains simple and does not introduce the **conceptual distinctions** or **contrary structures** that arise in philosophical reasoning.
>
> RepublicQA complements these datasets by providing **abstract conceptual semantics** and **rich contrary relationships**, enabling evaluations that engage both semantic and logical complexity.
>
> #### (2) Relation to synthetic logic datasets with adjustable complexity
> Synthetic pipelines (e.g., Qi et al., ICLR 2025) successfully scale **logical depth**, and we evaluate on their hardest configurations. However, they still operate in **factual, determinate domains**, and their **contrary-constructibility remains low** (≈0.13).
>
> Thus, while these datasets advance logical complexity, they do not supply the **semantically grounded contrary patterns** that motivate LogicAgent’s semiotic–logical integration.
>
>
> ---
> We sincerely thank the reviewer for the careful reading and thoughtful feedback. Your comments helped us sharpen the scope of our work and articulate its core contribution more clearly. We appreciate the opportunity to improve the manuscript through your suggestions. If there are any remaining concerns or points that would benefit from further clarification, we would be more than happy to address them.

---

### Official Review · Reviewer_oiWA · 2025-11-01

**Soundness:** 3
**Presentation:** 3
**Contribution:** 3
**Rating:** 4
**Confidence:** 4

**Summary:**

This paper introduces LogicAgent, a multi-perspective reasoning framework grounded in the Semiotic Square, designed to address the challenges of logical reasoning in LLMs when confronted with semantic ambiguity and abstract propositions. The framework operates by performing parallel deductions in FOL on a proposition, its contrary, and its contradictory, leveraging a multi-stage reflective verification mechanism to resolve inconsistencies. The authors also introduce RepublicQA, a new benchmark for this task characterized by high difficulty and semantic complexity derived from philosophical texts, on which their method achieves state-of-the-art performance, significantly outperforming strong baselines.

**Strengths:**

- The core idea of integrating a structuralist semantic tool (the Semiotic Square) with symbolic logic to mitigate semantic ambiguity is novel and compelling.
- The contribution of a new, manually annotated benchmark (RepublicQA) to address the lack of semantic complexity in existing datasets is valuable to the community.

**Weaknesses:**

- The methodology section lacks formal rigor. The paper would be significantly strengthened by adding more precise mathematical statements or lemmas that detail the formal assumptions and boundary conditions required to migrate the semiotic square into classical FOL. It should include, for example, a formal definition of the "existential import check" and its application.

- Reproducibility remains a concern. While the prompts are provided, the authors should add more concrete examples of side-by-side NL-to-FOL mappings. It is especially important to include complex cases involving nested quantifiers and negations, as these are critical for replicating the "Translator" module.

- The necessity of the *full* four-point Greimas Square is questionable. The authors should provide a targeted ablation study comparing the full four-point structure against a simpler three-point structure---one using S1, not S1, S2---to justify the framework's complexity.

- Some related works about LLM-based logical reasoning are missing, which should be compared with the proposed method or discussed on their difference. e.g.:

[1] Cumulative Reasoning with Large Language Models

[2] DetermLR: Augmenting LLM-based Logical Reasoning from Indeterminacy to Determinacy

**Questions:**

See above.

---

> ### Author Response · Authors · 2025-11-18
> **Rebuttal to Reviewer oiWA**
>
> We thank the reviewer for the thoughtful and constructive feedback, and we are glad that the novelty of combining the Semiotic Square with symbolic logic and the contribution of the *RepublicQA* benchmark were recognized. Below we address each concern in detail.
>
> ---
>
> **W1. “The methodology lacks formal rigor; more precise mathematical statements are needed.”**
>
> Thank you for raising this important point. In the revised manuscript, **Section 3.2 now provides the formal foundation that governs the migration from the Semiotic Square into classical FOL**. This section makes explicit the assumptions that were previously implicit and supplies the mathematical conditions necessary for sound symbolic reasoning.
>
> Specifically, Section 3.2 introduces:
>
> - **A formal definition of the existential-import check (Definition 1)**, which specifies when a proposition is non-vacuous under a given premise set and therefore meaningful within classical semantics.
> - **A lemma on valid contrary pairs (Lemma 1)**, which states the conditions under which semiotically generated propositions qualify as genuine contraries in FOL: both must satisfy existential import and must be jointly unsatisfiable across all models of the premises.
> ---
>
> **W2. Reproducibility and need for more NL→FOL examples.**
> We appreciate the reviewer’s focus on reproducibility, which is central to LogicAgent.
>
> **(1) Reproducibility.**
> Appendix E already includes step-wise case studies, and the supplementary material now provides the full translator code. For the camera-ready version we will open-source all code and datasets, ensuring that every NL→FOL mapping is fully verifiable end-to-end.
>
> **(2) Complex examples.**
> In response to the reviewer’s suggestion, the revision now includes **Appendix Table 7**, which provides *parallel NL→FOL mappings* across simple, medium, and hard items for all four datasets. These examples cover nested quantifiers, multi-clause negation, exclusive-or constructions, and cyclic dependencies, demonstrating how the translator handles scoping, operator normalization, and CFG-constrained linearization. To further illustrate end-to-end behavior, **Appendix E** presents a detailed case study showing how these mappings interact with the planner–solver pipeline and reflective verification, reinforcing the transparency and rigor of LogicAgent’s dual-form, multi-perspective reasoning process.
>
> ---
> **W3. Necessity of the Full Four-Point Greimas Square**
>
> We appreciate the reviewer’s question regarding whether the full four-point Semiotic Square is necessary. To directly address this, we conducted an ablation comparing the full structure against progressively simplified three-point variants. The results are shown below.
>
> **Table: Ablation on Greimas-Square Variants**
> | Setting              | FOLIO | RepublicQA* | Avg   |
> |----------------------|-------|-------------|-------|
> | S1                   | 72.06 | 76.50       | 74.28 |
> | S1 + ¬S1             | 76.47 | 78.50       | 77.49 |
> | S1 + ¬S1 + S2        | 79.41 | 81.50       | 80.46 |
> | **Full 4-point (Ours)** | **79.90** | **82.50** | **81.20** |
>
> Two observations emerge:
>
> 1. **The full four-point structure consistently yields the strongest performance**, confirming that the contrary axis (S2 vs ¬S2) is not redundant but functionally important.
> 2. The incremental gain from ¬S2 is smaller because only a subset of questions reach this stage of the reasoning cascade, specifically those that contain genuine contrary semantics and exhibit sufficient logical depth. This pattern is consistent with our design: LogicAgent is intended for propositions where semantic ambiguity and logical complexity naturally co-occur.
>
> Thus, the ablation supports the reviewer’s intuition while showing that the full Semiotic Square is essential for high-ambiguity, high-complexity philosophical reasoning tasks.

---

> ### Author Response · Authors · 2025-11-18
>
> **W4. On missing related works**
>
> We thank the reviewer for highlighting these two relevant lines of work. We have added both to the camera-ready and briefly clarify their relation to LogicAgent below.
>
> **Cumulative Reasoning (CR).**
> CR constructs a Proposer–Verifier–Reporter DAG to increase the arrival probability of correct multi-step reasoning. This aligns with our broad motivation of reducing error propagation, and we will cite CR accordingly. Importantly, CR operates *within a single semantic commitment*, whereas LogicAgent addresses a different failure mode: ambiguity arising from competing interpretations of the same proposition. Our multi-perspective design (S1 / ¬S1 / S2 / ¬S2) complements rather than substitutes CR by resolving semantic underdetermination before symbolic deduction begins.
>
> **DetermLR.**
> DetermLR focuses on converting indeterminate reasoning into deterministic chains via calibrated consistency checks. We view it as orthogonal to our contribution: DetermLR assumes a fixed logical framing, while LogicAgent explicitly *structures* the space of possible framings using the Semiotic Square and parallel FOL verification. DetermLR enhances chain reliability; LogicAgent enhances *conceptual grounding* when propositions admit multiple contrary or contradictory readings.
>
> We have added both works to the introduction and related-work sections.
>
> ---
>
> We appreciate the reviewer’s detailed suggestions. Collectively, these revisions make the paper more **formally rigorous** and **empirically transparent**, reinforcing *LogicAgent* as a theoretically grounded and reproducible framework for reasoning under semantic ambiguity. If further concerns remain about the scope or clarity of these updates, we would be glad to address them in the next revision.

---

> > ### Comment · Reviewer_oiWA · 2025-11-21
> >
> > Thanks for the authors' response. Some of my concerns are resolved. However, the effectiveness of the proposed method has not yet been sufficiently verified based on the limited baselines. I still believe it is necessary and beneficial to include more advanced LLM-based reasoning methods as your new comparisons (ToT, CR, DetermLR, etc), instead of discussing only in the related works. If the proposed method can be comparable to more recent baselines, I will reconsider adjusting the evaluation of the contribution of this work.

---

> ### Author Response · Authors · 2025-11-26
> **Supplementary baselines (1/2)**
>
> ### W4 – Supplementary baselines (ToT, CR, DetermLR, etc.)
>
> We thank the reviewer for this constructive suggestion. Our initial reply updated the related work but did not add new baselines because we were still reproducing ToT and CR. In the revision, we have now incorporated **ToT** (Yao et al., 2023) and **CR** (Zhang et al., 2023) across all datasets (Table). They are now fully integrated in the revised results.
>
> **Performance comparison across RepublicQA and other logical reasoning benchmarks**
> **Bold = best**, *italic = second-best*
>
> | Type     | Method        | **RepublicQA** |           |          | **Others** |            |         |             |          |
> |----------|---------------|----------------|-----------|----------|-----------------------|------------|---------|-------------|----------|
> |          |               | Qwen2.5 ↑      | GPT-4o ↑  | Avg ↑    | Pronto ↑             | ProofW. ↑ | FOLIO ↑ | Prover. ↑ | Avg ↑    |
> | **LR**   | Naive         | 68.50          | 74.00     | 71.25    | 82.00                 | 59.17      | 60.29   | 39.60       | 60.27    |
> |          | CoT           | 72.00          | 75.00     | 73.50    | 92.40                 | 63.17      | 68.42   | 47.20       | 67.80    |
> | **AR**   | CR            | 57.00          | 71.00     | 64.00    | 80.20                 | 58.33      | 71.57   | 51.80       | 65.48    |
> |          | ToT           | 56.00          | 69.50     | 62.75    | 82.50                 | 46.67      | 72.06   | 53.40       | 63.66    |
> | **SR**   | Logic-LM      | 70.00          | 73.50     | 71.75    | 91.89                 | 63.82      | *71.93* | *62.40*     | *72.51*  |
> |          | SymbCoT       | *76.00*        | 80.50     | 78.25    | *95.20*               | *64.67*    | 70.59   | 57.20       | 71.92    |
> | **AR+SR**| Aristotle      | 74.50          | *82.50*   | *78.50*  | 94.80                 | 63.23      | 68.68   | 56.20       | 70.73    |
> |          | **LogicAgent**| **82.50 (+6.50)** | **87.00 (+4.50)** | **84.75 (+6.25)** | **97.80 (+2.60)** | **71.95 (+7.28)** | **79.90 (+7.97)** | **68.60 (+6.20)** | **79.56 (+7.05)** |
>
> #### (1) Why ToT and CR were not included initially
> Our first version categorized prior methods into three groups:  **Prompt-based**, **Symbolic/FOL-based**, and **Hybrid architectures**.  After revisiting the reviewer’s concern, and following the taxonomy of *Aristotle* (Xu et al., 2024), we adopt a clearer and more principled classification:
>
> 1. **Linear Reasoning (LR)**
> 2. **Aggregative Reasoning (AR)** – multi-path exploration + aggregation (e.g., ToT, CR)
> 3. **Symbolic Reasoning (SR)**
> 4. **AR+SR** – hybrid symbolic–aggregative pipelines (e.g., *Aristotle*, LogicAgent)
>
> This made clear that our original baselines indeed lacked representatives of **AR**, and we now include ToT and CR accordingly.
>
> #### (2) Why AR baselines underperform on ProofWriter and other benchmarks
> Despite their popularity, AR methods show structural failure modes in logical tasks, particularly those requiring full negation chains such as ProofWriter, where correct contradiction detection depends on completing the entire negative reasoning path.
>
> - **Reasoning myopia under negation.**
>   CR performs prediction+voting; ToT performs BFS+voting.  Both generate many partial chains but rarely maintain **complete negation chains**, which are essential for detecting *False* in datasets like ProofWriter.  Incomplete negative paths default to *Uncertain*, driving accuracy down.
>
> - **Attention dilution with longer searches.**
>   BFS-style exploration in ToT produces long, noisy contexts.  With fixed context windows, relevant inferential states lose attention weight; the model drifts toward default “uncertain” or surface-level heuristics.
>
> - **Majority vote amplifies uncertainty.**
>   When mid-depth chains are uncertain (common for negation or implicit premises), voting collapses to *Uncertain*, even if a valid chain exists.  As a result, **Naive** (single-chain inference) often outperforms AR methods because it forces the model to commit rather than average over noisy partial chains.

---

> > ### Author Response · Authors · 2025-11-26
> > **Supplementary baselines (2/2)**
> >
> > #### (3) Why AR baselines fail even more severely on RepublicQA
> >
> > RepublicQA uniquely couples **semantic complexity** with **logical depth**.
> > This combination stresses AR methods in two ways:
> >
> > 1. **Semantic abstraction breaks majority voting.**
> >    Philosophical propositions often require interpreting conceptual distinctions (e.g., justice vs. power).
> >    Multi-path exploration cannot maintain these distinctions coherently, leading to semantically inconsistent partial chains and high *Uncertain* rates.
> >
> > 2. **Contrary structures require perspective-sensitive reasoning.**
> >    RepublicQA includes non-trivial contraries (≈70%), which demand reasoning across *multiple semantic interpretations*. AR methods explore paths blindly without modeling these semantic perspectives; thus, their aggregated conclusions lose the necessary structure for correct evaluation.
> >
> > 3. **Higher semantic difficulty increases chain brittleness.**
> >
> > | **Hop** | **FOLIO Acc.** | **FOLIO FK** | **ProofWriter Acc.** | **PW FK** | **ProverQA Acc.** | **ProvQA FK** | **RepublicQA Acc.** | **RQA FK** |
> > |--------:|----------------:|--------------:|-----------------------:|-----------:|---------------------:|----------------:|----------------------:|-------------:|
> > | 4 | 0.5161 | 6.45 | 0.6029 | 1.65 | 0.5636 | 8.27 | 0.4286 | 12.99 |
> > | 5 | 0.4474 | 7.72 | 0.4634 | 1.60 | 0.4231 | 8.40 | 0.3125 | 14.68 |
> > | 6 | 0.3750 | 7.35 | 0.4000 | 1.38 | 0.2500 | 7.40 | 0.0000 | 17.70 |
> >
> >    As FKGL rises from ~13→18 with hop depth, AR chains become longer and more semantically noisy.  Voting over brittle chains magnifies noise, causing systematic underperformance in exactly the settings RepublicQA is designed to probe. Because LogicAgent reasons over structured semantic perspectives (S₁, S₂, ¬S₁, ¬S₂) before deduction, it avoids the inconsistent partial-chain effects that undermine AR pipelines.
> >
> >
> > **Inference Time Comparison (s/item)**
> >
> > | Method                | ProofWriter | FOLIO   | RepublicQA | ProverQA | Avg    |
> > |-----------------------|------------:|--------:|-----------:|---------:|--------:|
> > | **Cumulative Reasoning** | 203.34     | 204.60 | 195.31     | 221.60   | 206.21 |
> > | **Tree-of-Thoughts**     | 232.64     | 194.89 | 208.64     | 294.58   | 232.69 |
> > | **Ours**                 | 162.05     | 203.84 | 186.05     | 261.20   | 203.29 |
> >
> > We also report an inference-time comparison. Both CR and ToT require substantially more time per item because they rely on wide multi-path exploration, yet their accuracy remains noticeably weaker. Their core limitation is that these methods aggregate many trajectories without integrating symbolic structure. As a result, the search frequently drifts and often collapses into Uncertain. In contrast, LogicAgent incorporates symbolic constraints and verification at each reasoning path, which improves path correctness while maintaining competitive runtime.
> >
> > ###
> >
> > We sincerely appreciate the reviewer’s continued engagement. Your suggestions directly motivated the expanded experimental suite, which we believe strengthens the empirical grounding and overall positioning of our work.

---

> > > ### Comment · Reviewer_oiWA · 2025-11-27
> > >
> > > Thanks for the authors' response and efforts. I have no further questions and decide to update the evaluation. Looking forward to seeing these new interesting analysis results and discussions in the paper.

---

> > > > ### Author Response · Authors · 2025-11-28
> > > >
> > > > Thank you again for your continued engagement and thoughtful review.
> > > > Your positive feedback is greatly encouraging to us.
> > > > We have now incorporated the updated results into the revision and addressed the corresponding analyses and discussions accordingly.
> > > >
> > > > Best regards,
> > > >
> > > > The authors

---

### Official Review · Reviewer_7b1v · 2025-11-03

**Soundness:** 2
**Presentation:** 3
**Contribution:** 2
**Rating:** 2
**Confidence:** 4

**Summary:**

The paper proposes LogicAgent, a reasoning framework that marries Greimas’ Semiotic Square with first-order logic (FOL), adds an existential-import check to avoid vacuous truths, and evaluates propositions with a three-valued scheme {True, False, Uncertain}.

**Strengths:**

- Provide a new dataset RepublicQA with college-level difficulty.
- Propose LogicAgent with great result on benchmarks.

**Weaknesses:**

- Dataset Scale: The size of the newly dataset RepublicQA is too small (n=200). This limited scale raises concerns about the statistical robustness of the findings and the dataset's general utility.

- Benchmark Reporting: The results on "Other Benchmarks" are reported as an aggregate average. Could the authors provide a detailed, disaggregated breakdown of performance for each individual benchmark (e.g., ProntoQA, ProofWriter, FOLIO, and ProverQA)?

- Dataset Generalizability: The decision to construct the RepublicQA dataset exclusively from a single source, Plato's "Republic," is questionable. This narrow domain scope inherently limits the dataset's diversity and generalizability.

- Novelty of Methodology: using Greimas’ Semiotic Square and extending the evaluation space from a binary {True, False} to a three-valued scheme {True, False, Uncertain} appear to lack significant innovation. As far as I know, many existing works, particularly in probabilistic logic, have implemented similar "de-binarization" approaches to handle uncertainty. The authors need to better justify the novelty of their method against this prior art.

**Questions:**

deliver my issues in the weakness's sections

---

> ### Author Response · Authors · 2025-11-18
> **Rebuttal to Reviewer 7b1v**
>
> We sincerely thank the reviewer for the time spent evaluating our paper and for recognizing the strengths of **LogicAgent**, including its introduction of the *RepublicQA* dataset and its strong benchmark results.  However, several key points in this review appear to stem from misunderstandings of the paper’s scope, structure, and contributions. We address these below in detail.
>
> ---
>
> ### **W1 & W3 — Dataset Scale and Generalizability**
>
> **RepublicQA is designed for semantic depth, not size.**
> Its compactness reflects the difficulty of constructing items that require *abstract semantic interpretation* and *symbolic reasoning*. As shown in **revised Appendix Table 6**, RepublicQA reaches an FKGL of **11.94**, which is a **42% increase** over the strongest prior benchmark (ProverQA, 8.44). Each item requires extracting a philosophical proposition, constructing its contrary and contradictory, and aligning all forms under first-order logic with an existential-import check. This semantic structuring is absent in ProntoQA, ProofWriter, FOLIO, and ProverQA, where semantic content remains relatively static even as logical depth increases.
>
> **RepublicQA uniquely couples semantic and logical complexity.**
> As shown in revised Table 4, existing benchmarks maintain almost constant semantic difficulty across hops—for example, FOLIO’s FKGL ranges **6.45–7.72**, and ProverQA’s FKGL ranges **7.40–8.40**—indicating that increasing logical depth does not increase semantic complexity.
> RepublicQA is the only dataset in which semantic difficulty increases together with logical depth: FKGL rises from **12.99 to 17.70** (4–6 hops), while accuracy simultaneously decreases from **0.4286 to 0.0000**, producing a tightly coupled semantic–logical difficulty regime absent in prior work. This coupling is the core reason RepublicQA provides value beyond dataset size.
>
> | **Hop** | **FOLIO Acc.** | **FOLIO FK** | **ProofWriter Acc.** | **PW FK** | **ProverQA Acc.** | **ProvQA FK** | **RepublicQA Acc.** | **RQA FK** |
> |--------:|----------------:|--------------:|-----------------------:|-----------:|---------------------:|----------------:|----------------------:|-------------:|
> | 4 | 0.5161 | 6.45 | 0.6029 | 1.65 | 0.5636 | 8.27 | 0.4286 | 12.99 |
> | 5 | 0.4474 | 7.72 | 0.4634 | 1.60 | 0.4231 | 8.40 | 0.3125 | 14.68 |
> | 6 | 0.3750 | 7.35 | 0.4000 | 1.38 | 0.2500 | 7.40 | 0.0000 | 17.70 |
>
> **Expansion to 600 items with multi-source philosophical grounding.**
> The revised Appendix D expands RepublicQA using additional philosophical sources—including *Aristotle’s Nicomachean Ethics*—following the multi-source construction strategy of Qi et al., ICLR 2025. **Appendix Table 6** provides the full comparison across basic statistics and semantic-complexity measures, showing that RepublicQA consistently exceeds prior benchmarks in lexical diversity, conceptual abstraction, and contrary structure. Revised Section 4 emphasizes that the benchmark is designed for *semantic richness and conceptual contrast*, not scale, and serves as a diagnostic test for human-level conceptual distinctions rather than a broad corpus.
>
> | **Benchmark** | **Total** | **Vocab** | **Logic steps** |
> |---------------|-----------|-----------|-----------------|
> | ProntoQA      | 500       | 69        | 11              |
> | FOLIO         | 204      | 1,021     | 0*              |
> | ProofWriter   | 600       | 61        | 0*              |
> | ProverQA      | 500       | 2,453     | 25.73           |
> | RepublicQA    | 600        | 4,070     | 16.22           |
>
> **Complementarity rather than narrowness.**
> Although its textual sources are philosophical, RepublicQA expands the reasoning space covered by symbolic logic benchmarks by introducing structured semantic contraries and ambiguity—phenomena missing from synthetic or deterministically generated datasets. Through its Semiotic–Logical integration, it offers a complementary evaluation axis that existing benchmarks cannot provide.
>
>
> ---
>
> ### **W2. “Benchmark Reporting: Only aggregate results are provided.”**
>
> **Clarification of misunderstanding.**
> While the **abstract and introduction** report the *average performance* for brevity, **Section 5.2** and **Table 2** already provide a **per-benchmark breakdown**—*ProntoQA* (+2.60), *ProofWriter* (+7.28), *FOLIO* (+7.97), and *ProverQA* (+6.20)—showing an **average improvement of 7.05 points** over the strongest baseline.
>
> **Consistency with established reporting standards.**
> This presentation mirrors the conventions of recent reasoning frameworks such as *Symbolic CoT* (Xu et al., 2024) and *Aristotle* (Xu et al., 2024), which likewise report **per-dataset metrics** followed by an overall mean for readability.

---

> ### Author Response · Authors · 2025-11-18
>
> ### **W4. “Novelty: The three-valued {True, False, Uncertain} evaluation is not innovative.”**
>
>
> **The contribution is not the third label but a mechanism that makes uncertainty logically and semantically distinguishable.**
> Existing three-valued systems (for example, Symbolic CoT by Xu et al., 2024) operate only on the original proposition \(S_1\). When \(S_1\) cannot be entailed, the model often returns “False,” which makes non-entailment and genuine indeterminacy indistinguishable. This limitation stems from reasoning over a single proposition without considering alternative interpretations.
>
> **LogicAgent resolves this by bringing semantic variation into the reasoning space.**
> We evaluate four propositions \(S₁, S₂, ¬S₁, ¬S₂\), which represent distinct semantic perspectives. Each is processed through the planner–solver deduction pipeline, and their outcomes are compared in the reflective stage.
> Uncertainty arises only when none of the four propositions is entailed and their cross-checks do not resolve a conclusion. This allows LogicAgent to separate contradiction (*False*) from indeterminacy (*Uncertain*) in a principled way.
>
> **The key novelty is the integration of semantic perspectives with formal deduction.**
> Earlier datasets and methods focus almost exclusively on logical depth, while LogicAgent jointly models semantic interpretation and symbolic reasoning. This unified treatment brings multiple meanings and their logical consequences into the inference process, enabling forms of analysis that are not possible when operating on a single proposition alone.
>
> ---
> We thank you for the thoughtful and constructive evaluation. Your comments helped us articulate more clearly that RepublicQA is positioned as a semantic–logical diagnostic benchmark rather than a training-scale dataset. If there are further concerns about our work, we welcome additional guidance and will be glad to refine the revision accordingly.

---

> > ### Comment · Reviewer_7b1v · 2025-11-20
> >
> > Thank you for your reply. I remain contain about the dataset size and data source. It seems that almost all data stems from Greek philosophy, which will constrain to the limited domain.
> > But given that most of my questions are solved, I raised my score.

---

> > > ### Author Response · Authors · 2025-11-20
> > >
> > > Thank you sincerely for your thoughtful follow-up and for raising your score. We truly appreciate your careful reading of our work and the time you invested in helping us refine it. We are actively considering your concern regarding dataset scope and are exploring broader philosophical and conceptual sources for future extensions.
> > >
> > > Best regards,
> > > The Authors

---

### Note · Authors · 2025-12-25

I have read and agree with the venue's withdrawal policy on behalf of myself and my co-authors.